**Combining Ensemble Kalman Filter and Reservoir Computing to**
**predict spatio-temporal chaotic systems from imperfect observations**
**and models**
**Futo Tomizawa[1] and Yohei Sawada[1,2,3]**
[1]School of Engineering, the University of Tokyo, Tokyo, Japan
[2]Meteorological Research Institute, Japan Meteorological Agency, Tsukuba, Japan
[3]RIKEN Center for Computational Science, Kobe, Japan
**Abstract**
Prediction of spatio-temporal chaotic systems is important in various fields, such as Numerical
Weather Prediction (NWP). While data assimilation methods have been applied in NWP, machine
learning techniques, such as Reservoir Computing (RC), are recently recognized as promising tools to
predict spatio-temporal chaotic systems. However, the sensitivity of the skill of the machine learning
based prediction to the imperfectness of observations is unclear. In this study, we evaluate the skill of
RC with noisy and sparsely distributed observations. We intensively compare the performances of RC
and Local Ensemble Transform Kalman Filter (LETKF) by applying them to the prediction of the
Lorenz 96 system. In order to increase the scalability to larger systems, we applied parallelized RC
framework. Although RC can successfully predict the Lorenz 96 system if the system is perfectly
observed, we find that RC is vulnerable to observation sparsity compared with LETKF. To overcome
this limitation of RC, we propose to combine LETKF and RC. In our proposed method, the system is
predicted by RC that learned the analysis time series estimated by LETKF. Our proposed method can
successfully predict the Lorenz 96 system using noisy and sparsely distributed observations. Most
importantly, our method can predict better than LETKF when the process-based model is imperfect.
**1. Introduction**
In Numerical Weather Prediction (NWP), it is required to obtain the optimal estimation of atmospheric
state variables by observations and process-based models which are both imperfect. Observations of
atmospheric states are sparse and noisy, and numerical models inevitably include biases. In addition,
models used in NWP are known to be chaotic, which makes the prediction substantially difficult. To
accurately predict the future atmospheric state, it is important to develop methods to predict spatio-
tempral chaotic dynamical systems from imperfect observations and models.
Traditionally, data assimilation methods have been widely used in geosciences and NWP systems.
Data assimilation is a generic term of approaches to estimate the state from observations and model
outputs based on their errors. The state estimated by data assimilation is used as the initial value, and
the future state is predicted by models alone. Data assimilation is currently adopted in operational
NWP systems. Many data assimilation frameworks have been proposed, e.g. 4D variational methods
(4D-VAR; Bannister, 2017), Ensemble Kalman Filter (EnKF; Houtekamer & Zhang, 2016), or their
derivatives, and they have been applied to many kinds of weather prediction tasks, such as the
prediction of short-term rainfall events (e.g. Sawada et al., 2019; Yokota et al., 2018), and severe
storms (e.g. Zhang et al., 2016). Although data assimilation can efficiently estimate the unobservable
state variables from noisy observations, the prediction skill is degraded if the model has large biases.

On the other hand, model-free prediction methods based on machine learning have received much
attention recently. In the context of dynamical system theory, previous works have developed the
methods to reproduce the dynamics by inferring it purely from observation data (Rajendra and
Brahmajirao, 2020), or by combining a data-driven approach and physical knowledge on the systems
(Karniadakis et al., 2021). In the NWP context, many previous studies have successfully applied
machine learning to predict chaotic dynamics. Vlachas et al. (2018) successfully applied Long-Short
Term Memory (LSTM; Hochreiter & Schmidhuber, 1997) to predict the dynamics of the Lorenz96
model, Kuramoto-Sivashinski Equation, and the barotropic climate model which is a simple
atmospheric circulation model. Asanjan et al. (2018) showed that LSTM can accurately predict the
future precipitation by learning satellite observation data. Nguyen & Bae (2020) successfully applied
LSTM to generate area-averaged precipitation prediction for hydrological forecasting.

In addition to LSTM, Reservoir Computing (RC), which was first introduced by Jaeger & Haas (2004),
has been found to be suitable to predict spatio-temporal chaotic systems. Pathak et al. (2017)
successfully applied RC to predict the dynamics of Lorenz equation and Kuramoto-Sivashinski
Equation. Lu et al. (2017) showed that RC can be used to estimate state variables from sparse
observations if the whole system was perfectly observed as training data. (Pathak et al., 2018b)
succeeded in using a parallelized RC to predict each segment of the state space locally, which enhanced
the scalability of RC to much higher dimensional systems. Chattopadhyay et al. (2020) revealed that
RC can predict the dynamics of the Lorenz 96 model more accurately than LSTM and Artificial Neural
Network (ANN). In addition to the accuracy, RC also has an advantage in computational costs. RC
can learn the dynamics only by training a single matrix as a linear minimization problem just once,
while other neural networks have to train numerous parameters and need many iterations (Lu et al.,
2017). Thanks to this feature, the computational costs needed to train RC is cheaper than LSTM and
ANN.

However, Vlachas et al. (2020) revealed that the prediction accuracy of RC is degraded when all of
the state variables cannot be observed. It can be a serious problem since the observation sparsity is
often the case in geosciences and the NWP systems. Brajard et al. (2020) pointed out this issue and
successfully trained the Convolutional Neural Network with sparse observations, by combining with
EnKF. However, their method needs to iterate the data assimilation and training until the prediction
accuracy of the trained model converges. Although one can stop the iteration in a few times, it can be
longer, and the training can be computationally expensive if one should wait until the convergence.
Bocquet et al. (2020) proposed a method to combine EnKF and machine learning methods to obtain
both the state estimation and the surrogate model online. They showed successful results without using
the process-based model at all. Dueben & Bauer (2018) mentioned that the spatio-temporal
heterogeneity of observation data made it difficult to train machine learning models, and they
suggested to use the model or reanalysis as training data. Weyn et al. (2019) successfully trained
machine learning models using the atmospheric reanalysis data.

We aim to propose the novel methodology to predict spatio-temporal chaotic systems from imperfect
observations and models. First, we reveal the limitation of the stand-alone use of RC under realistic
situations (i.e., imperfect observations and models). Then, we propose a new method to maximize the
potential of RC to predict chaotic systems from imperfect models and observations, which is even
computationally feasible. As Dueben & Bauer (2018) proposed, we make RC learn the analysis data
series generated by a data assimilation method. Our new method can accurately predict from imperfect
observations. Most importantly, we found that our proposed method is more robust to model biases
than the stand-alone use of data assimilation methods.


**2.  Methods**
**2.1  Lorenz 96 model and OSSE**
We used a low dimensional spatio-temporal chaotic model, the Lorenz 96 model (L96), to perform
experiments with various parameter settings. L96 is a model introduced by Lorenz & Emanuel (1998)
and has been commonly used in data assimilation studies (e.g. Kotsuki et al., 2017; Miyoshi, 2005;
Penny, 2014; Raboudi et al., 2018). L96 is recognized as a good testbed for the operational NWP
problems (Penny, 2014).

In this model, we consider a ring structured and $m$ dimensional discrete state space $x_1, x_2, \ldots, x_m$
(that is, $x_m$ is adjacent to $x_1$), and define the system dynamics as follows:
$$\frac{dx_i}{dt} = (x_{i+1} - x_{i-2}) x_{i-1} - x_i + F \qquad (1)$$
where $F$ stands for the force parameter, and we define $x_{-1} = x_{m-1}$, $x_0 = x_m$, and $x_{m+1} = x_1$.
Each term of this equation corresponds to advection, damping and forcing respectively. It is known
that the model with $m = 40$ and $F = 8$ shows chaotic dynamics with 13 positive Lyapunov
exponents (Lorenz and Emanuel, 1998), and this setting is commonly used in the previous studies (e.g.
Kotsuki et al., 2017; Miyoshi, 2005; Penny, 2014; Raboudi et al., 2018). The time width $\Delta t = 0.2$
corresponds to one day in real atmospheric motion from the view of dissipative decay time (Lorenz
and Emanuel, 1998).

As we use this conceptual model, we cannot obtain any observational data or "true" phenomena that
correspond to the model. Instead, we adopted Observing System Simulation Experiment (OSSE). We
first prepared a time series by integrating equation (1) and regarded it as the "true" dynamics (called
Nature Run). Observation data can be calculated from this time series adding some perturbation:
$$\boldsymbol{y^O} = \boldsymbol{Hx} + \boldsymbol{\epsilon} \tag{2}$$
where $\boldsymbol{y^O} \in \mathbb{R}^h$ is the observation value, $\boldsymbol{H}$ is the $m \times h$ observation matrix, $\boldsymbol{\epsilon} \in \mathbb{R}^h$ is the
observational error whose each element is independent and identically distributed from a Gaussian
distribution $N(0,\ e)$ for observation error $e$.

In each experiment, the form of L96 used to generate Nature Run is unknown, and the model used to
make prediction can be different from that for Nature Run. The difference between the model used for
Nature Run and that used for prediction corresponds to the model's bias in the context of NWP.

**2.2 Local Ensemble Transform Kalman Filter**
We used the Local Ensemble Transform Kalman Filter (LETKF, Hunt et al., 2007) as the data
assimilation method in this study. LETKF is one of the ensemble-based data assimilation methods,
which is considered to be applicable to the NWP problems in many previous studies (Sawada et al.,
2019; Yokota et al., 2018). LETKF is also used for the operational NWP in some countries (e.g.
Germany; Schraff et al., 2016).

LETKF and the family of ensemble Kalman filters have two steps; forecast and analysis. The analysis
step makes the state estimation based on the forecast variables and observations. The forecast step
makes the prediction from the current analysis variables to the time for the next analysis using the
model. The interval for each analysis is called "assimilation window".
Considering the stochastic error in the model, system dynamics can be represented as follows
(hereafter the subscript $k$ stands for the variable at time $k$, and the time width between $k$ and $k +$
1 corresponds to the assimilation window):
$$x_k^f = \mathcal{M}(x_{k-1}^a) + \eta_k, \qquad \eta_k \sim N(\mathbf{0}, \mathbf{Q}) \qquad (3)$$
where $x_k^f \in \mathbb{R}^m$ is the forecast variables, $x_{k-1}^a \in \mathbb{R}^m$ is the analysis variables, $\mathcal{M}: \mathbb{R}^m \to \mathbb{R}^m$ is
the model dynamics operator, $\boldsymbol{\eta} \in \mathbb{R}^m$ is the stochastic error and $N(\boldsymbol{0}, \boldsymbol{Q})$ means the multivariate
Gaussian distribution with mean $\boldsymbol{0}$ and $n \times n$ covariance matrix $\boldsymbol{Q}$. Since the error in the model is
assumed to follow the Gaussian distribution, forecasted state $x^f$ can also be considered as a random
variable from the Gaussian distribution. When the assimilation window is short, the Gaussian nature
of the forecast variables is preserved even if the model dynamics is nonlinear. In this situation, the
probability distribution of $x^f$ (and also $x^a$) can be parametrized by mean $\overline{x^f}$ ($\overline{x_k^a}$) and covariance
matrix $\boldsymbol{P}^f$ ($\boldsymbol{P}_k^a$).
Using the computed state vector $x_k^f$, observation variables can be estimated as follows:

$$y_k^f = \mathcal{H}\left(x_k^f\right) + \epsilon_k, \qquad \epsilon_k \sim N(\boldsymbol{0}, \boldsymbol{R}) \tag{4}$$

where $y^f \in \mathbb{R}^h$ is the estimated observation value, $\mathcal{H}: \mathbb{R}^m \to \mathbb{R}^h$ is the observation operator and
$\epsilon \in \mathbb{R}^h$ is the observation error sampled from $N(\boldsymbol{0}, \boldsymbol{R})$. Although $\mathcal{H}$ can be either linear or
nonlinear, we assume it to be linear in this study and expressed as a $h \times m$ matrix $\boldsymbol{H}$ (the treatment
of the nonlinear case is discussed in Hunt et al., 2007).

LETKF uses an ensemble of state variables to estimate the evolution of $\overline{x_k^f}$ and $\boldsymbol{P}_k^f$. The time
evolution of each ensemble members is as follows:

$$x_k^{f,(i)} = \mathcal{M}\left(x_{k-1}^{a,(i)}\right) \tag{5}$$

where $x_k^{f,(i)}$ is the $i$th ensemble member of forecast value at time $k$. Then the mean and covariance
of state variables can be expressed as follows:
$$\overline{x_k^f} \approx \frac{1}{N_e} \sum_{i=1}^{N_e} x_k^{f,(i)}, \qquad P_k^f = \frac{1}{N_e - 1} X_k^f \left( X_k^f \right)^T \tag{6}$$

where $N_e$ is the number of ensemble members and $X_k^f$ is the matrix whose $i$th column is the
deviation of the $i$th ensemble member from the ensemble mean.

In the analysis step, LETKF assimilates only the observations close to each grid point. Therefore, the
assimilated observations are different at different grid points and the analysis variables of each grid
points are computed separately.
For each grid points, observations to be assimilated are chosen. The rows or elements of $y^o$, $H$, and
$R$ corresponding to non-assimilated observations should be removed as the localization procedure.
"Smooth localization" can also be performed by multiplying some factors to each row of $R$ based on
the distance between target grid point and observation points (Hunt et al., 2007).
From the forecast ensemble, the mean and the covariance of the analysis ensemble can be calculated
in the ensemble subspace as follows:
$$\overline{w_k^a} = \tilde{P}_k^a \left( H X_k^f \right)^T R^{-1} \left( y^o - H \overline{x_k^f} \right)$$
$$\tilde{P}_f^a = \left[ (k-1)I + \left( H X_k^f \right)^T R^{-1} H X_k^f \right]^{-1} \tag{7}$$

where $w_k^a, \widetilde{P}_f^a$ stands for the mean and covariance of the analysis ensemble calculated in the ensemble
subspace. They can be transformed into model space as follows:
$$\overline{x_k^a} = \overline{x_k^f} + X_k^f \overline{w_k^a}$$

$$P_k^a = X_k^f \widetilde{P}_k^a \left(X_k^f\right)^T \tag{8}$$

On the other hand, as equation (6), we can consider the analysis covariance as the product of the
analysis ensemble matrix:
$$P_k^a = \frac{1}{N_e - 1} X_k^a (X_k^a)^T \tag{9}$$

where $X_k^a$ is the matrix whose $i$th column is the variation of the $i$th ensemble member from the
mean for the analysis ensemble. Therefore, decomposing $\widetilde{P}_k^a$ of equation (7) into square root, we can
get each analysis ensemble member at time $k$ without explicitly computing the covariance matrix in
the state space:
$$W_k^a (W_k^a)^T = \widetilde{P}_k^a, \qquad x_k^a = \overline{x_k^f} + \sqrt{N_e - 1} \, X_k^f w_k^a \tag{10}$$

where $w_k^a$ is the $i$th column of $W_k^a$ in the first equation. A covariance inflation parameter is
multiplied to take measures for the tendency of data assimilation to underestimate the uncertainty of
state estimate by empirically accounting for model noise (see equation (3)). See Hunt et al. (2007) for
more detailed derivation. Now, we can return to the equation (5) and iterate forecast and analysis step.

As in the real application, we consider the situation that the observations are not available in the
prediction period. Predictions are made by the model alone, using the latest analysis state variables as
the initial condition:

$$x^f_{K+1} = \widetilde{\mathcal{M}}(\overline{x^a_K}), \;\; x^f_{K+2} = \widetilde{\mathcal{M}}(x^f_{K+1}), \;\; ... \tag{11}$$

where $x^f_k$ is the prediction variables at time $k$, $\widetilde{\mathcal{M}}$ is the prediction model (an imperfect L96 model),
and $\overline{x^a_K}$ is the mean of the analysis ensemble at the initial time of the prediction. This way of making
prediction is called "Extended Forecast", and we call this prediction "LETKF-Ext" in this study, to
distinguish it from the forecast-analysis iteration of LETKF.

**2.3 Reservoir Computing**
**2.3.1    Description of Reservoir Computing Architecture**
We use Reservoir Computing (RC) as the machine learning framework. RC is a type of Recurrent
Neural Network, which has a single hidden layer called reservoir. Figure 1 shows its architecture. As
mentioned in Section 1, previous works have shown that RC can predict the dynamics of spatio-
temporal chaotic systems.

The state of the reservoir layer at timestep $k$ is represented as a vector $r_k \in \mathbb{R}^{D_r}$, which evolves
given the input vector $u_k \in \mathbb{R}^m$ as follows:

$$r_{k+1} = \tanh[Ar_k + W_{in}u_k] \tag{12}$$

where $W_{in}$ is the $D_r \times m$ input matrix which maps the input vector to the reservoir space, and $A$
is the $D_r \times D_r$ adjacency matrix of the reservoir which determines the reservoir dynamics. $W_{in}$
should be determined to have only one nonzero component in each row, and each nonzero component
is sampled from uniform distribution of $[-a, a]$ for some parameter $a$. $A$ has a proportion of $d$
nonzero components with random values from uniform distribution, and it is normalized to have the
maximum eigenvalue $\rho$. The reservoir size $D_r$ should be determined based on the size of the state
space. From the reservoir state, we can compute the output vector $v$ as follows:

$$v_k = W_{out} f(r_k) \tag{13}$$

where $W_{out}$ is the $M \times D_r$ output matrix which maps the reservoir state to the state space, and
$f: \mathbb{R}^{D_r} \to \mathbb{R}^{D_r}$ is an operator of nonlinear transformation. The nonlinear transformation is essential
for the accurate prediction (Chattopadhyay et al., 2020). It is important that $A$ and $W_{in}$ are fixed and
only $W_{out}$ will be trained by just solving a linear problem. Therefore, the computational cost required
to train RC is small and it is an outstanding advantage of RC compared to the other neural network
frameworks.

In the training phase, we set the switch in the Figure 1 to the training configuration. Given a training
data series $\{u_0, u_1, \ldots, u_K\}$, we can generate the reservoir state series $\{r_1, r_2, \ldots, r_{K+1}\}$ by equation
(12). By using the training data and reservoir state series, we can determine the $W_{out}$ matrix by ridge
regression. We minimize the following square error function with respect to $\boldsymbol{W_{out}}$:
$$\sum_{i=1}^{n}\|\boldsymbol{u_k} - \boldsymbol{W_{out}}\boldsymbol{f}(\boldsymbol{r_k})\|^2 + \beta \cdot trace(\boldsymbol{W_{out}}\boldsymbol{W_{out}^T}) \tag{14}$$

where $\|\boldsymbol{x}\| = x^T x$ and $\beta$ is the ridge regression parameter (normally a small positive number).
Since the objective function (14) is quadratic, it is differentiable. The optimal value can be obtained
by just solving a linear equation as follows:
$$\boldsymbol{W_{out}} = \boldsymbol{U}\boldsymbol{R}^T \, (\boldsymbol{R}\boldsymbol{R}^T + \beta \boldsymbol{I})^{-1} \tag{15}$$

where $\boldsymbol{I}$ is the $D_r \times D_r$ identity matrix and $\boldsymbol{R}, \boldsymbol{U}$ are the matrices whose $kth$ column is the vector
$\boldsymbol{f}(\boldsymbol{r_k}), \boldsymbol{u_k}$, respectively.

Then, we can shift to the predicting phase. Before we predict with the network, we first need to "spin
up" the reservoir state. The spin up process was done by giving the time series before the initial value
$\{\boldsymbol{u_{-k}}, \boldsymbol{u_{-k+1}}, \dots, \boldsymbol{u_{-1}}\}$ to the network and calculate the reservoir state right before the beginning of the
prediction via equation (12). After that, the output layer is connected to the input layer, and the network
becomes recursive. In this configuration, the output value $\boldsymbol{v_k}$ of equation (13) is used as the next
input value $\boldsymbol{u_k}$ of equation (12). Once we give the initial value $\boldsymbol{u_0}$, the network will iterate equation
(12) and (13) spontaneously, and the prediction will be yielded. At this point, RC can now be used as
the surrogate model that mimics the state dynamics:
$$\boldsymbol{x}_{k+1}^f = \widetilde{\mathcal{M}}_{RC}\left(\boldsymbol{x}_k^f, \{\boldsymbol{x}_k^{train}\}_{1 \leq k \leq K}\right) \tag{16}$$

where $x_k^f$ is the prediction variables at time $k$, $\widetilde{\mathcal{M}}_{RC}$ is the dynamics of RC (equations (12) and
(13)) and $\left\{x_k^{train}\right\}_{1\leq k\leq K} = \{x_1^{train}, x_2^{train}, \dots, x_K^{train}\}$ is the time series of training data.

Considering the real application, it is natural to assume that the observation data can only be used as
the training data and the initial value for the RC prediction. In this paper we call this type of prediction
"RC-Obs". Prediction time series here can be expressed using equation (16) as follows:

$$x_{K+1}^f = \widetilde{\mathcal{M}}_{RC}\left(y_K^O, \{y_k^O\}_{1\leq k\leq K}\right), \quad x_{K+2}^f = \widetilde{\mathcal{M}}_{RC}\left(x_{K+1}^f, \{y_k^O\}_{1\leq k\leq K}\right), \dots \tag{17}$$

where $\{y_k^O\}_{1\leq k\leq K} = \{y_1^O, y_2^O, \dots\}$ is the observation time series and $y_K^O$ is the observation at the initial
time of the prediction. As in equation (14), input and output of RC must be in the same space.
Therefore, in this case, prediction variables $x_k^f$ has the same dimensionality as $y_k^O$, and the non-
observable grid points are not predicted by this prediction scheme.

**2.3.2    Parallelized Reservoir Computing**
In general, the required reservoir size $D_r$ for accurate prediction increases as the dimension of the
state space $m$ increases. Since the RC framework needs to keep adjacency matrix $\boldsymbol{A}$ on the memory,
and to perform inverse matrix calculation of $D_r \times D_r$ matrix (equation (15)), too large reservoir size
leads to unfeasible computational cost. (Pathak et al., 2018b) proposed a solution to this issue, which
is called the parallelized reservoir approach.
In this approach, the state space is divided into $g$ groups, all of which contains $q = m/g$ state
variables:

$$g_k^{(i)} = \left(u_{k,(i-1)\times q+1}, u_{k,(i-1)\times q+2}, \ldots, u_{k,i\times q}\right)^T, i = 1, 2, \ldots, g \tag{18}$$

where $g_k^{(i)}$ is the $i$th group at time $k$, $u_{k,j}$ is the $j$th state variable at time $k$. Each group is
predicted by different reservoir placed in parallel. $i$th reservoir accepts the state variables of $i$th
group as well as adjacent $l$ grids, which can be expressed as follows:

$$h_k^{(i)} = \left(u_{k,(i-1)\times q+1-l}, u_{k,(i-1)\times q+2-l}, \ldots, u_{k,i\times q+l}\right)^T \tag{19}$$

where $h_k^{(i)}$ is the input vector for $i$th reservoir at time $k$. The dynamics of each reservoir can be
expressed as follows according to equation (12):

$$r_{k+1}^{(i)} = \tanh\left[A^{(i)}r_k^{(i)} + W_{in}^{(i)}h_k^{(i)}\right] \tag{20}$$

where $r_k^{(i)}$, $A^{(i)}$, $W_{in}^{(i)}$ and $W_{out}^{(i)}$ are the reservoir state vector, adjacency matrix input matrix, and
output matrix for $i$th reservoir. Each reservoir is trained independently using equation (13) so that:

$$g_k^{(i)} = W_{out}^{(i)}f\left(r_k^{(i)}\right) \tag{21}$$

where $W_{out}^{(i)}$ is the output matrix in the $i$th reservoir. The prediction scheme of parallelized RC is
summarized in Figure 2. The strategy of parallelization is similar to the localization of data
assimilation. As LETKF ignores correlations between distant grid points, parallelized reservoir
computing assumes that the state variable of a grid point at the next time step depends only on the
state variables of neighboring points. In contrast, ordinary RC assumes that the time evolution at one
grid point is affected by all points in the state space, which may be inefficient in many applications in
geoscience such as NWP.

**2.4 Combination of RC and LETKF**
As discussed so far and we will quantitatively discuss in the section 4, LETKF-Ext and RC-Obs have
contrasting advantages and disadvantages. LETKF-Ext can accurately predict even if the observation
is noisy and/or sparsely distributed, while RC-Obs is vulnerable to the imperfectness in observation.
On the other hand, LETKF-Ext can be strongly affected by the model biases since the prediction of
LETKF-Ext depends only on the model after obtaining the initial condition, while RC-Obs has no
dependence to the accuracy of the model as it only uses the observation data for training and prediction.

Therefore, the combination of LETKF and RC has a potential to push the limit of these two individual
prediction methods and realize accurate and robust prediction. The weakness of RC-Obs is that we
can only use the observational data directly, which is inevitably sparse in the real application, although
RC is vulnerable to this imperfectness. In our proposed method, we make RC learn the analysis time
series generated by LETKF instead of directly learning observation data.

Suppose we have sparse and noisy observations for the training data. If we take observations as inputs
and analysis variables as outputs, LETKF can be considered as an operator to estimate the full state
variables from the sparse observations:

$$\{\overline{x_k^a}\}_{1 \leq k \leq K} = \{\mathcal{D}(y_k^O)\}_{1 \leq k \leq K} \tag{22}$$

where $\{\overline{x_k^a}\}_{1 \leq k \leq K} = \{x_1^a, x_2^a, ..., x_K^a\}$ is the full-state variables (time series of the LETKF analysis
ensemble mean), $y_k^O$ is the observation, and $\mathcal{D}: \mathbb{R}^n \to \mathbb{R}^m$ represents the state estimation operator,
which is realized by LETKF in this study. Then, RC is trained by using $\{x_k^a\}_{1 \leq k \leq K}$ as the training
data set. In this way, RC can mimic the dynamics of analysis time series computed by forecast-analysis
cycle of LETKF. Prediction can be generated by using the analysis variables at current time step ($x_K^a$)
as the initial value. Since RC is trained with LETKF analysis variables, we call this method "RC-Anl".
By using the notation of equation (16), the prediction of RC-Anl can be expressed as follows:

$$x_{K+1}^f = \widetilde{\mathcal{M}}_{RC}(x_K^a, \{x_k^a\}_{1 \leq k \leq K}), \qquad x_{K+2}^f = \widetilde{\mathcal{M}}_{RC}(x_{K+1}^f, \{x_k^a\}_{1 \leq k \leq K}), ... \tag{23}$$

where $\{x_k^a\}_{1 \leq k \leq K} = \{x_1^a, x_2^a, ..., x_K^a\}$ is the time series of the LETKF analysis variables. The
schematics of the LETKF-Ext, RC-Obs, and RC-Anl are shown in the figure 3. Initial values and
model dynamics used in each method are compared in Table 1.

Our proposed combination method is expected to predict more accurately than RC-Obs since the
training data always exist in all the grid points, even if the observation is sparse. Also, especially if the
model is substantially biased, the analysis time series generated by LETKF is more accurate than the
model output itself. It means that RC-Anl is expected to be able to predict more accurately than
LETKF-Ext.

**3. Experiment Design**
To generate the Nature Run, L96 with $m = 40$, $F = 8$ was used, and it was numerically integrated
by 4th order Runge-Kutta method by time width $\Delta t = 0.005$. Before calculating the Nature Run, the
L96 equation was integrated for 1440000 timesteps for spin up. In the following experiment, the $F$
term in the model was changed to represent the model bias.

Here, we assume that the source of the model bias is unknown. When the source of bias is only the
uncertainty in model parameters, and uncertain parameters which significantly induce the model bias
are completely identified, optimization methods can estimate the value of the uncertain parameters to
minimize the gaps between simulation and observation. This problem can also be solved by data
assimilation methods (e.g. Bocquet and Sakov, 2013). However, it is difficult to calibrate the model
when the source of uncertainty is unknown. Our proposed method does not need to identify the source
of model bias so that it may be useful especially when the source of model bias is unknown. This is
often the case in the large and complex model such as NWP systems.

The setting for LETKF was based on Miyoshi and Yamane (2007). As the localization process, the
observation points within 10 indices are chosen to be assimilated for every grid point. The "smooth
localization" is also performed on observation covariance $\boldsymbol{R}$. Since we assume that each observation
error is independent and thus $\boldsymbol{R}$ is diagonal, the localization procedure can be done just by dividing
each diagonal elements of observation covariance $\boldsymbol{R}$ by the value $w$ calculated as follows:

349
$$w(r) = \exp\left(-\frac{r^2}{18}\right) \tag{24}$$

where $r$ is the distance between each observation point and each analyzed point. For every grid point,
the observation point with $w(r) \geq 0.0001$ are chosen to be assimilated. In equation (10), a
"covariance inflation factor", which was set to 1.05 in our study, was multiplied to $\widetilde{\boldsymbol{P}}_k^a$ in each
iteration to maintain the sufficiently large background error covariance by empirically accounting for
model noise (see equation (3)). Ensemble size $N_e$ was set to 20.

The parameter values of parallelized RC used in this study is similar to Vlachas et al. (2020), but was
slightly modified. Parameter settings used in the RC experiments are shown in Table 1. Jiang and Lai
(2019) revealed that the performance of RC is sensitive to $\rho$ and it needs to be tuned. We identified
the proper value of $\rho$ by sensitivity studies. Other parameters do not substantially affect the
prediction accuracy, and we selected them based on the settings in previous works such as Vlachas et
al. (2020). The nonlinear transformation function for the output layer in equation (13) is the same as
Chattopadhyay et al. (2020), which is represented as follows:
$$f(r_i) = \begin{cases} r_i & (i \text{ is odd}) \\ r_{i-1} \times r_{i-2} & (i \text{ is even}) \end{cases} \tag{25}$$

where $r_i$ is the $i$th element of $\boldsymbol{r}$. Note that the form of the transformation function can be flexible;
one can use a different form of the function to predict Lorenz 96 (Chattopadhyay et al., 2020), or the
same function can be used to predict other systems (Pathak et al., 2017). In the prediction phase, we
used the data for 100 timesteps before the prediction initial time for the reservoir spin up.

We implement numerical experiments to investigate the performance of RC-Obs, LETKF-Ext and
RC-Anl to predict L96 dynamics. First, we evaluate the performance of RC-Obs under perfect
observations (all the grid points are observed with no error) and quantify the effect of the observation
imperfectness (i.e. observation error and spatio-temporal sparsity), to investigate the prediction skill
of the stand-alone use of RC and LETKF. Second, we evaluate the performance of RC-Anl. We
investigate the performance of RC-Anl and LETKF-Ext as the functions of the observation density
and model biases.

In each experiment, we prepare 200000 timesteps of Nature Run. The first 100000 timesteps are used
for the training of RC or for the spinning up of LETKF, and the rest of them are used for the evaluation
of each method. Every prediction is repeated 100 times to avoid the effect of the heterogeneity of data.
For the LETKF-Ext prediction, the analysis time series of all the evaluation data is firstly generated.
Then, the analysis variables for one every 1000 timestep is taken as the initial conditions and total 100
prediction runs are performed. For the RC-Obs prediction, evaluation data are equally divided into
100 sets and the prediction is identically done for each set. For the RC-Anl prediction, the analysis
time series of training data are used for training, and the prediction is performed using the same initial
condition as LETKF-Ext. Each prediction set of LETKF-Ext, RC-Obs, and RC-Anl corresponds to
the same time range.

The prediction accuracy of each method is evaluated by taking the average of RMSE of 100 sets for
each timestep. We call this metric mean RMSE (*mRMSE*), and can be represented as follows:
$$mRMSE(t) = \frac{1}{100} \sum_{i=1}^{100} \sqrt{\frac{1}{m} \sum_{j=1}^{m} \left( u_j^{(i)}(t) - x_j^{(i)}(t) \right)^2} \tag{26}$$

where $t$ is the number of the steps elapsed from the prediction initial time, $x_j^{(i)}(t)$ is the $j$th nodal
value of the $i$th prediction set at time $t$ and $u_j^{(i)}(t)$ is the corresponding value of Nature Run. Using
this metric, we can see how the prediction accuracy is degraded as time elapses from initial time (so
called "forecast lead time").

**4. Results**
Figure 4 shows the Hovmöller diagram of a prediction of RC-Obs and Nature Run. Figure 4 also
shows the difference between prediction and Nature Run, as well as the actual prediction results so
that we can see how long we can keep the prediction accurate. RC is trained with perfect observation
($e = 0$ at all grid point). Figure 4 shows that RC-Obs predicts accurately within approximately 200
timesteps.

Figure 5 shows the time variation of the *mRMSE* (see equation (26)) of RC-Obs with perfect
observation. It also shows that RC-Obs can predict with good accuracy for approximately 200
timesteps. It should be noted that LETKF (as well as other data assimilation methods) just replaces
the model's forecast with the initial conditions identical to Nature Run when all state variables can be
perfectly observed, and thus the prediction accuracy of LETKF-Ext will be perfect if we have no
model bias. LETKF-Ext is much superior to RC-Obs under this regime (not shown).

Next, we evaluated the sensitivity of the prediction skill of both LETKF-Ext and RC-Obs to the
imperfectness of the observations. Figure 6a and 6b show the effect of the observation error on the
prediction skill. The value of observation error $e$ is changed from 0.1 to 1.5 and the *mRMSE* time
series is drawn. We can see that LETKF-Ext is more sensitive to the increase of observation error than
RC-Obs, although the LETKF-Ext is superior in accuracy to RC-Obs within this range of observation
error.

However, RC-Obs showed a greater sensitivity to the density of observation points than LETKF-Ext.
Figures 7a and 7b show the sensitivity of the prediction accuracy of LETKF-Ext and RC-Obs,
respectively, to the number of observed grid points. Observation is reduced as uniformly as possible.
The observation network in each experiment is shown in Table 2. Even though we can observe a small
part of the system, the accuracy of LETKF-Ext changed only slightly. On the other hand, the accuracy
of RC-Obs gets worse when we remove a few observations. As assumed in the section 2.4, we verified
that RC-Obs is more sensitive to the observation sparsity than LETKF-Ext.

We tested the prediction skill of our newly proposed method, RC-Anl, under perfect models and sparse
observations. Here, we used the observation error $e = 1.0$. Figure 8 shows the change of the *mRMSE*
time series of RC-Anl with the different number of observed grid points. It indicates that the
vulnerability of the prediction accuracy to the change of the number of observed grid points, which is
found in RC-Obs, no longer exists in RC-Anl. Although the prediction accuracy is lower than LETKF-
Ext (Figure 7a), our new method indicates a robustness to the observation sparsity and overcomes the
limitation of the stand-alone RC.

Moreover, when the model used in LETKF is biased, RC-Anl outperforms LETKF-Ext. Figures 9a
and 9b show the change of the *mRMSE* time series when changing the model biases. The number of
the observed points was set to 20. The $F$ term in equation (1) was changed from the true value 8 (the
$F$ value of the model for Nature Run) to values in $[5.0, 11.0]$ as the model bias, and the accuracy of
LETKF-Ext and RC-Anl is plotted. The accuracy of LETKF-Ext was slightly better than that of RC-
Anl when the model was not biased ($F = 8$; green line). However, when the bias is large (e.g. $F =$
10; gray line), RC-Anl showed the better prediction accuracy.

We confirmed this result by comparing the *mRMSE* value of RC-Anl and LETKF-Ext at the specific
forecast lead-time. Figure 10 shows the value of $mRMSE(80)$ (see equation (26)) as the function of
the value of the $F$ term. Both two lines that show the skill of RC-Anl (blue) and LETKF-Ext (red)
are convex downward and have a minimum at $F = 8$, meaning that the accuracy of both prediction
methods are the best when the model is not biased. In addition, as long as $F$ value is in the interval
$[7.5, 8.5]$, LETKF-Ext has the better accuracy than RC-Anl. However, if the model bias become larger
than that, RC-Anl becomes more accurate than LETKF-Ext. As the bias increases, the difference
between the $mRMSE(80)$ of two methods becomes larger, and the superiority of RC-Anl becomes
more obvious. We found that RC-Anl can predict more accurately than LETKF-Ext when the model
is biased.

We also checked the robustness for the training data size. Figure 11 shows the change of the accuracy
of RC-Anl by changing the size of training data from 100000 to 10000 timesteps. We confirmed that
the prediction accuracy did not change until the size was reduced to 25000 timesteps. Although we
have used a large size of training data (100000 timesteps; 68 model years) so far, the results are robust
to the reduction of the size of the training data.

**5.  Discussion**

By comparing the prediction skill of RC-Obs and LETKF-Ext, we confirmed that RC-Obs can predict
with accuracy comparable to LETKF-Ext, if we have perfect observations. This result is consistent
with Chattopadhyay et al. (2020), Pathak et al. (2017), and Vlachas et al. (2020), and we can expect
that RC has a potential to predict various kinds of spatio-temporal chaotic systems.

However, Vlachas et al. (2020) revealed that the prediction accuracy of RC is substantially degraded
when the observed grid points are reduced, compared to other machine learning techniques such as
LSTM. Our result is consistent with their study. In contrast, Chattopadhyay et al. (2020) showed that
RC can predict the multi-scale chaotic system correctly even though only the largest scale dynamics
is observed. Comparing these results, we can suggest that the states in the scale of dominant dynamics
should be observed almost perfectly to accurately predict the future state by RC.

Therefore, when we use RC to predict spatio-temporal chaotic systems with sparse observation data,
we need to interpolate them to generate the appropriate training data. However, the interpolated data
inevitably includes errors even if the observation data itself has no error, so it should be verified that
RC can predict accurately by training data with some errors. Previous works such as Chattopadhyay
et al. (2020), Pathak et al. (2017), or Vlachas et al. (2020) have not considered the impact of error in
the training data. We found that the prediction accuracy of RC degrades as the error in training data
grows, but the degradation rate is not so large (if all the training data of all the grid points are obtained).
We can expect from this result that RC trained with the interpolated observation data can predict
accurately to some extent, but the interpolated data should be as accurate as possible.

In this study, LETKF was used to prepare the training data for RC, since LETKF can interpolate the
observations and reduce their error at the same time. We showed that our proposed approach correctly
works. Brajard et al. (2020) also made Convolutional Neural Network (CNN) learn the dynamics from
sparse observation data and successfully predict the dynamics of the L96 model. However, as
mentioned in the introduction section, Brajard et al. (2020) iterated the learning and data assimilation
until they converge, because it replaced the model used in data assimilation with CNN. Although their
model-free method has an advantage that it was not affected by the process-based model's
reproducibility of the phenomena, it can be computationally expensive since the number of iterates
can be relatively large. By contrast, we need to train RC just one time, because we use the process-
based model (i.e. data assimilation method) to prepare the training data. We overcome the problem of
computational feasibility.

Note also that the computational cost to train RC is much cheaper than the other neural networks.
Since the framework of our method does not depend on a specific machine learning framework, we
believe that we can flexibly choose other machine learning methods such as RNN, LSTM, ANN, etc.
Previous studies such as Chattopadhyay et al. (2020) or Vlachas et al. (2020) revealed that these
methods show competitive performances compared to RC in predicting spatio-temporal chaos. Using
them instead of RC in our method would probably give similar results. However, the advantage of RC
is its cheap training procedure. RC does not need to perform an expensive back-propagation method
for training, unlike other neural networks (Chattopadhyay et al., 2020; Lu et al., 2017). Therefore, RC
is considered as a promising tool for predicting spatio-temporal chaos. Although our method has
flexibility in the choice of machine learning methods, we consider that the good performance with RC
is important in this research context.

The good performance of our proposed method supports the suggestion of Dueben & Bauer (2018),
in which machine learning should be applied to the analysis data generated by data assimilation
methods as the first step of the application of machine learning to weather prediction. As Weyn et al.
(2019) did, we successfully trained the machine learning model with the analysis data.

Most importantly, we also found that the prediction by RC-Anl is more robust to the model biases than
the extended forecast by LETKF (i.e. LETKF-Ext). This result suggests that our method can be
beneficial in various real problems, as the model in real applications inevitably contains some biases.
Pathak et al. (2018a) developed the hybrid prediction system of RC and a biased model. Although
Pathak et al. (2018a) successfully predicted the spatio-temporal chaotic systems using the biased
models, they needed perfect observations to train their RC. The advantage of our proposed method
compared to these RC studies is that we allow both models and observation networks to be imperfect.
As in the review by Karniadakis et al. (2021), methodologies to train the dynamics from noisy
observational data by integrating data and physical knowledge are attracting attentions. In the NWP
context, some studies proposed methods to combine data assimilation and machine learning to emulate
the system dynamics from imperfect model and observations (e.g. Bocquet et al., 2019, 2020; Brajard
et al., 2020; Dueben and Bauer, 2018), and these approaches are getting popular. Our study
significantly contributes to this emerging research field.

Although we tested our method only on 40-dimensional Lorenz 96 system, (Pathak et al., 2018b)
indicated that parallelized RC can be extended to predict the dynamics of substantially high
dimensional chaos such as 200-dimensional Kuramoto-Sivashinski equation with small computational
costs. Moreover, the applicability to the realistic NWP problems has also been discussed by their
sequel study (Wikner et al., 2020). These studies imply It implies that the findings of this study can
also be applied to higher dimensional systems.

In NWP problems, it is often the case that homogenous observation data of high resolution are not
available over a wide range of time and space, which can be an obstacle to applying machine learning
to NWP tasks (Dueben & Bauer, 2018). We revealed that RC is robust for the temporal sparsity of
observations, and RC can be trained with relatively small training data sets.

However, since the Lorenz 96 model (and other conceptual models such as Kuramoto-Sivashinski
equation) is ergodic, it is unclear that our method can be applied to real NWP problems directly, which
are possibly non-ergodic. Although our proposed method has a potential to extend to larger and more
complex problems, further studies are needed.


## 6. Conclusion


The prediction skills of the extended forecast with LETKF (LETKF-Ext), RC that learned the
observation data (RC-Obs), and RC that learned the LETKF analysis data (RC-Anl) were evaluated
under imperfect models and observations, using the Lorenz 96 model. We found that the prediction by
RC-Obs is substantially vulnerable to the sparsity of the observation network. Our proposed method,
RC-Anl, can overcome this vulnerability. In addition, RC-Anl could predict more accurately than
LETKF-Ext when the process-based model is biased. Our new method is robust to the imperfectness
of both models and observations and we might obtain similar results higher dimensional and more
complexed systems. Further studies on more complicated models or operational atmospheric models
are expected.

**Code Availability**


The source code for RC and Lorenz96 model is available at:
https://doi.org/10.5281/zenodo.3907291, and for LETKF at:
https://github.com/takemasa-miyoshi/letkf

**Acknowledgement**


This work was supported by the Japan Society for the Promotion of Science KAKENHI grant
JP17K18352 and JP18H03800, the JAXA grant ER2GWF102, the JST AIP Grant JPMJCR19U2, and
the JST FOREST program. We thank four anonymous reviewers for their constructive comments.

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

**Table 1**. Parameter values of RC used in each experiment

| Parameter | Description | Value |
|---|---|---|
| $D_r$ | reservoir size | 2000 |
| $a$ | Input matrix scale | 0.5 |
| $d$ | adjacency matrix density | 0.005 |
| $\rho$ | adjacency matrix spectral radius | 1.0 |
| $\beta$ | ridge regression parameter | 0.0001 |
| $g$ | number of reservoir groups | 20 |
| $l$ | reservoir input overlaps | 4 |


653         **Table 2**. Summary of three prediction frameworks

| Name | Initial Value | Model for prediction |
|---|---|---|
| LETKF-Ext | LETKF analysis | the model used in LETKF |
| RC-Obs | observation | RC trained with observation |
| RC-Anl | LETKF analysis | RC trained with LETKF analysis |


655         **Table 3.** The indices of observed grid points.

| # Observed | Grid point index |
|---|---|
| | 1 2 3 4 5 6 7 8 9 10 11 12 13 14 15 16 17 18 19 20 21 22 23 24 25 26 27 28 29 30 31 32 33 34 35 36 37 38 39 40 |
| 40 | ● ● ● ● ● ● ● ● ● ● ● ● ● ● ● ● ● ● ● ● ● ● ● ● ● ● ● ● ● ● ● ● ● ● ● ● ● ● ● ● |
| 38 | ● ● ● ● ● ● ● ● ● ● ● ● ● ● ● ● ● ● ●  ● ● ● ● ● ● ● ● ● ● ● ● ● ● ● ● ● ● ● |
| 36 | ● ● ● ● ● ● ● ●  ● ● ● ● ● ● ● ● ●  ● ● ● ● ● ● ●  ● ● ● ● ● ● ● ● ● ● |
| 30 | ● ●  ● ●  ● ● ●  ● ● ●  ● ● ●  ● ● ●  ● ● ●  ● ● ●  ● ● ● |
| 20 | ● ●  ● ●  ● ●  ● ●  ● ●  ● ●  ● ●  ● ●  ● ● |


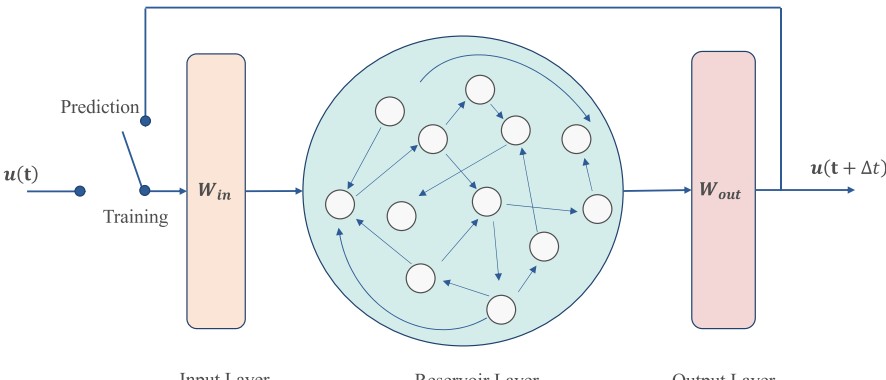

**Figure 1**. The conceptual diagram of reservoir computing architecture. The network consists of an
input layer, a hidden layer called reservoir, and an output layer.


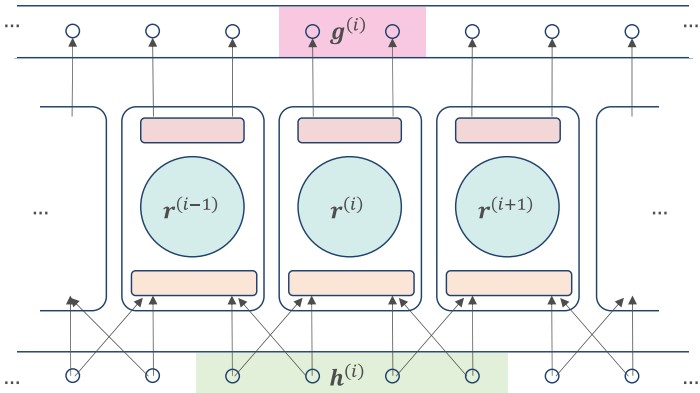

**Figure 2**. The conceptual diagram of parallelized reservoir computing architecture. The state space is
separated into some groups and the same number of reservoirs are put parallelly. Each reservoir groups
accepts the inputs from the corresponding group and some adjacent grids and predict the dynamics of
the corresponding group.

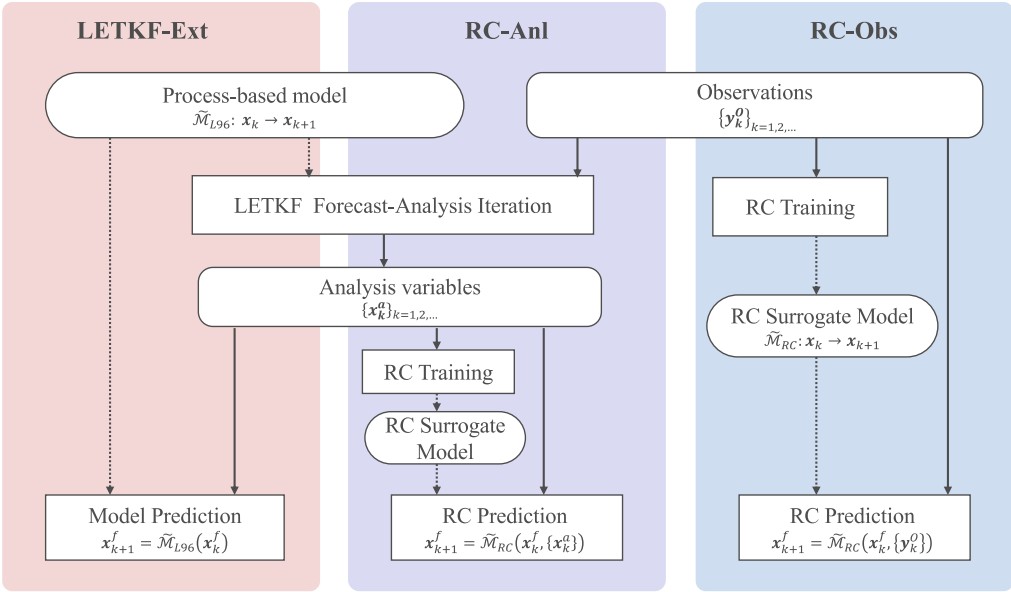

**Figure 3**. The algorithm flow of LETKF-Ext, RC-Anl, and RC-Obs. Solid and dotted lines show the
flow of variables and models (either process-based or data-driven surrogate), respectively.

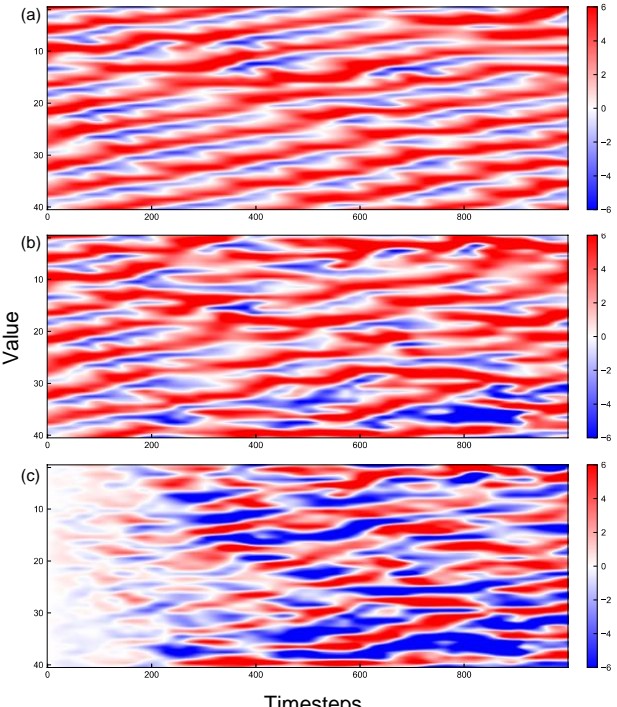

**Figure 4.** The Hovmöller diagram of (a) Nature Run, (b) A prediction of RC-Obs, (c) difference of (a)
and (b). Horizontal axis shows the timesteps and vertical axis shows the nodal number. Value at each
timestep and node is represented by the color.

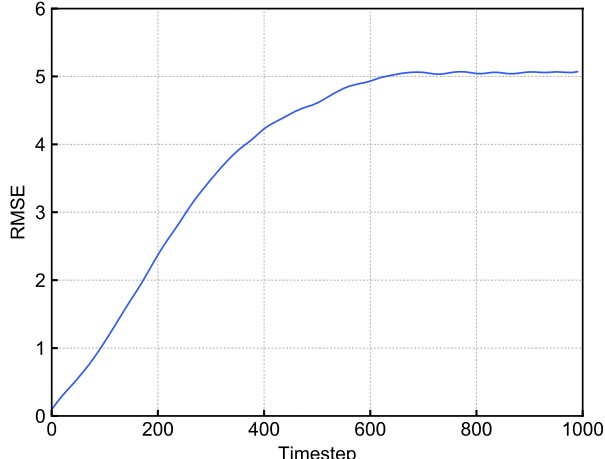

**Figure 5**. The *mRMSE* time series of the predictions of RC-Obs with perfect observation. Horizontal
axis shows the timestep and vertical shows the value of *mRMSE*.


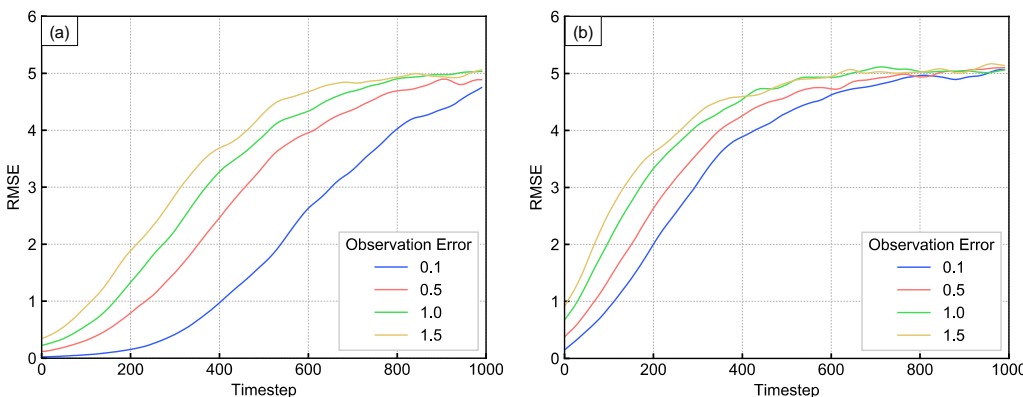

**Figure 6**. The *mRMSE* time series of the predictions of (a)LETKF-Ext and (b)RC-Obs with noisy
observation. Each color corresponds to the observation error indicated by the legend.


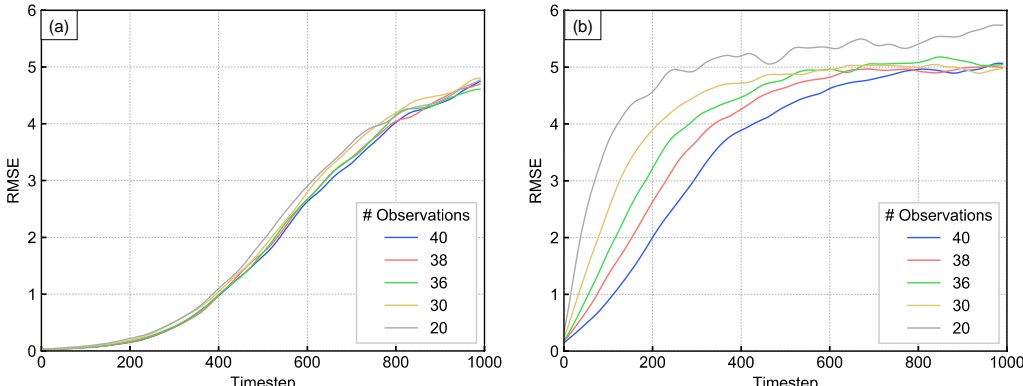

**Figure 7**. The *mRMSE* time series of the predictions of (a)LETKF-Ext and (b)RC-Obs with spatially
sparse observation. Each color corresponds to the number of the observation points.

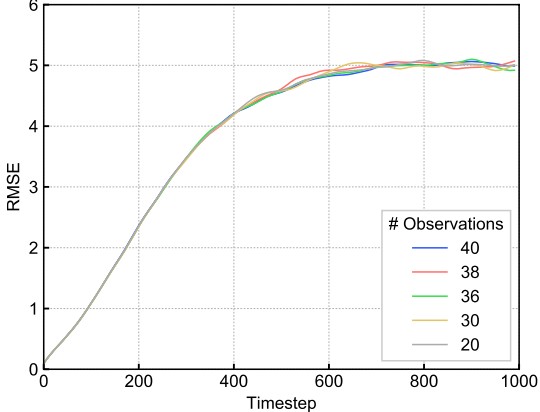

**Figure 8**. The same as figure4, for the RC-Anl prediction.


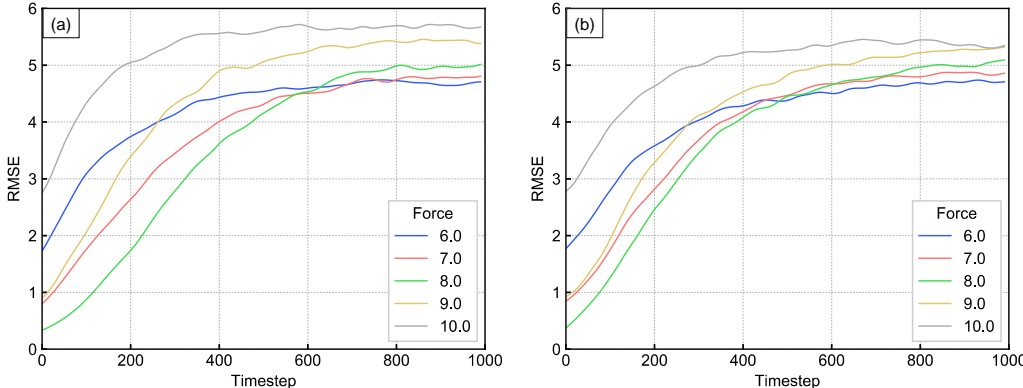

**Figure 9**. The *mRMSE* time series of the predictions of (a)LETKF-Ext and (b)RC-Anl with biased

model. Each color corresponds to each value of *F* term.


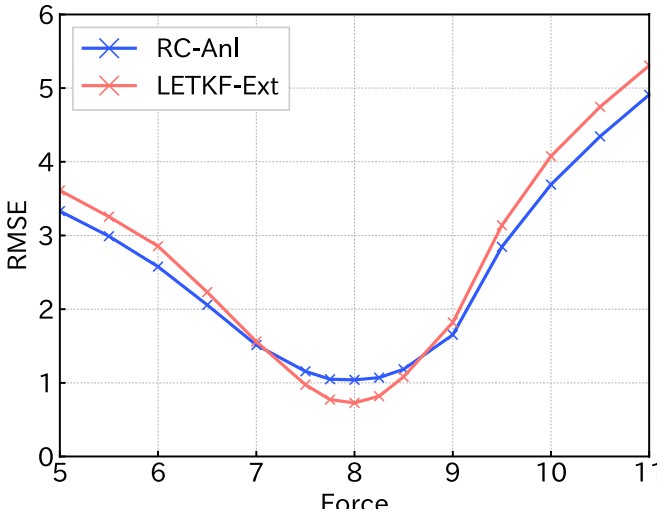

**Figure 10**. The *mRMSE(80)* of the predictions of LETKF-Ext(red) and RC-Anl(blue) for each model

bias. Horizontal axis shows the value of the force parameter of equation (1) (8 is the true value) and

vertical axis shows the value of *mRMSE*.

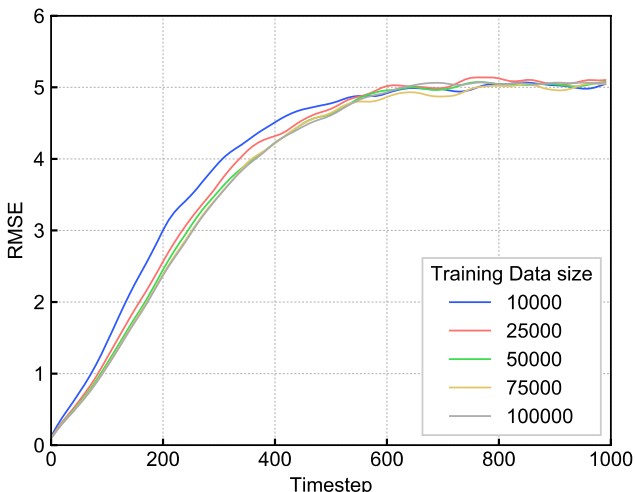

**Figure 11**. The *mRMSE* time series of the predictions of RC-Anl with various length of training data,
with perfect observation and perfect model. Each color corresponds to the value of the size of training
data.