# Peer review of "Combining Ensemble Kalman Filter and Reservoir Computing to predict spatio-temporal chaotic systems from imperfect observations and models"

_Geoscientific Model Development, 2020_

## Referee Comment (RC1) · Anonymous Referee #1 · 23 Aug 2020

Major comments:

Section 2.4 "Combination of RC and LETKF" is the scientific innovation that is at the core of the paper, however the authors only explore the idea in a purely textual form. Some equations comparing the three methods (RC-Obs, RC-Anl, LETKF) or perhaps a diagram would be immensely beneficial to the average reader, as that will be more eye-catching, and would help explain the papers innovation.

Section 3 "Experimental Design" mention the parameters used to construct the RC

network, however motivations for the choices are lacking, and would help potential follow-up works, including those by practitioners to intuit logical choices.

Equation (15) represents Gaussian kernel localization, however this is not explicitly mentioned, which seems very strange, as a reader familiar with data assimilation literature would be more familiar with that framing of the method. Additionally, such applications are typically done on the inverse of the matrix R (which is also explicitly taken to be diagonal), through a Woodbury matrix identity application, though such details are glossed over in the text.

The strangest thing is the choice of $m = 8$ for the experimental design. The standard L96 with $m = 40$ variables is known to be very chaotic and have 13 positive Lyapunov exponents. In that regime, a lot is known about the system. However as far as the reviewer can tell, the case of $m = 8$ has to be weakly chaotic by necessity. It would be of benefit to the casual reader (and the reviewer) to state some facts about this regime of the system.

The reviewer wonders if for such a small system it might have been of benefit to examine the more rigorously studied 3-variable Lorenz system, for which a lot is known, in terms of reservoir computing, data assimilation, and general nature of the system. The same type of sparsity experiments could be performed with such a system.

Section 4 "Results" contains a statement to the effect of "if 'e' is set to zero then the LETKF does not work". This is technically true, but very misleading. Take the Kalman filter formula with perfect and full observations $x^a = x^f - (x^f - y^o) = y^o$. Under the regime of perfect observation ($e = 0$, $H = I$), and Kalman filter would replace the analysis by the observations, and collapse the distribution of the uncertainty about the analysis. This means that in this regime, the LETKF would replace each ensemble member by the observations, even if a particular implementation would fail. It would make sense to—instead of using the LETKF—to simply look at the predictive regime of the RC-Obs. The $e = 0.01$ is entirely unnecessary.

The choice of which indices are observed in such experiments is not given as far as the reviewer can tell.

In section 5 "Discussion" the authors mention the generalization of their methodology to other dynamical systems. The authors focus on the generalization to "higher dimensional systems" as their main worry. The reviewer believes that another more substantial worry is that Lorenz '96 is known to be ergodic, while most systems used for NWP are possibly not, (the authors mention the KS system, but that is also possibly ergodic for well-posed initial conditions). It would be of benefit to mention other possible shortcoming on testing only on the Lorenz '96 system.

Technical comments:

L14: "are recently" to "have been recently"

L61, L65 "Neural Networks" (plural).

L70 "trained a"

L72 "toward real-world problems"

L91 "for operational"

L94 "ring structured" to "cyclic"

---

## Referee Comment (RC2) · Anonymous Referee #2 · 27 Aug 2020

The summary of existing work, novel directions, and discussion of findings is strong.

Some further questions: Does learning the LETFK analysis with a different method (e.g. vanilla feedforward network) give the same result? In other words, is there anything interesting about reservoir computing in particular in this context? Does learning the analysis from some other method work in the same way?

---

## Referee Comment (RC3) · Anonymous Referee #3 · 21 Sep 2020

**1   Major issues**

In spite of the obvious care taken to write this manuscript, it has quite a few issues, mainly but not only related to the presentation of the theory and to the choices made for the numerical experiments:

- I don't see the point in presenting the Kalman filter. Why not go straight to the introduction of the (L)ETKF? The introduction of the Kalman filter is really unnec-

essary.

- The introduction of the LETKF can be perfected (see a few points below).

- Localisation is never explicitly mentioned, which is at serious odds with the use of the LETKF.

- The description of the main algorithm is not very precise. I believe that it could be significantly improved.

- The statements about the applicability of the method in the discussion and the conclusion are much too strong, and should either be retracted or significantly mitigated.

- The numerical experiment choices show that localisation is totally unnecessary here which significantly undermines the manuscript. Using the ETKF would have yielded results at least as good. In particular any claim of applicability to higher dimensional models the authors make in the discussion and conclusion is seriously undermined by this issue (see specific points below). In this vein, it is rather disappointing that the authors did not apply the *local* RC (together with the LETKF) which would have been a significant added value to the paper. This is what I initially expected from the title and abstract of the paper.

- I believe that the English of the manuscript could be significantly improved.

As a consequence, I believe the manuscript requires major revisions before being acceptable for publication.

**2   Typos, remarks and suggestions, some related to the main issues:**

1. l.46: "have been receiving" ⟶ "have received"

2. l.69: Brajard et al. 2020a and Brajard et al. 2020b correspond to the same paper. You can safely remove Brajard et al. 2020a (which is the arXiv preprint of Brajard et al. 2020b). However, there is another 2020 paper from these authors which is directly relevant to your manuscript, see below in the references (Brajard et al., 2020b).

3. l.71-72: "However, their method needs to iterate the data assimilation and training, which is computationally expensive and infeasible toward the real-world problem.": Not really. Actually it depends on the number of iterations. One can use only one iteration for instance – this is actually what you do. Please mitigate your statement.

4. l.67-75: Bocquet et al. (2020) have recently proposed to use a local EnKF (LETKF for instance) coupled with machine learning. This must be cited since this is very relevant to your paper.

5. l.94-97: Your definition of the periodic nature of the L96 model is not precise enough, since $x_0$ or $x_{m+1}$ are not defined for instance. Please improve the wording of your definition.

6. l.100: The characteristic time of the L96 model has been discussed in the original paper by Lorenz and Emanuel (1998). If don't believe that you should cite Miyoshi et al. 2005 here in the text.

7. l.108: "and identically distributed on Gaussian distribution" ⟶ "and identically distributed from a Gaussian distribution"

8. l.119: "in some countries": this is too vague a statement; which ones for instance?

9. l.126: "and the time width of k corresponds to the assimilation window": This statement is not clear to me.

10. l.129: I suggest the change "and $N(\mathbf{0}, \mathbf{Q})$ means the Gaussian distribution" $\longrightarrow$ "and $N(\mathbf{0}, \mathbf{Q})$ means the multivariate Gaussian distribution" in order to contrast this definition with your previous definition for the univariate $N(0, \epsilon)$.

11. l.134: "extracted from" $\longrightarrow$ "sampled from"

12. l.136-141: It is not clear at this point why you would assume the model to be linear.

13. l.149-150: "If either the model operator M or observation operator H is nonlinear, we cannot directly use this method.": that is a bit of an exaggeration. If the model is mildly nonlinear, the extended Kalman filter can be used. Please reformulate.

14. l.153: "is EnKF. EnKF uses" $\longrightarrow$ "is the EnKF. The EnKF uses"

15. l.161: Your statement is misleading; the covariances matrices in state space are actually never explicitly computed in the (L)ETKF! Please correct this.

16. l.171, Eq.(10): The second equation is incomplete; $x_k^f$ is missing.

17. page 10: How come you don't ever mention localisation for the LETKF?

18. l.180: Why use the dedicated name "LETKF-Ext"? Forecasting from the analysis is just standard proceeding. I may have missed something here.

19. l.184: "shows the architecture." $\longrightarrow$ "shows its architecture."

20. l.185: "in the Section 1, the previous works" $\longrightarrow$ "in Section 1, previous works"

21. l.189: "$\mathbf{u}_k \in \mathbb{R}^m$": is it $m$ or $M$ as displayed in l.191?

22. l.194: "is extracted from uniform distribution" $\longrightarrow$ "is sampled from the uniform distribution"

23. l.197: I guess $\mathbf{v}$ should be bold, as a vector.

24. l.200: "is the operator for nonlinear transformation." ⟶ "is an operator of nonlinear transformations."

25. l.202: "Therefore, the computational cost required to train RC is small": You do not explain why! In particular you do not mention that the problem to optimise is linear (the loss function is quadratic).

26. l.213: "matrix" ⟶ "matrices"

27. l.240-241: It seems to me like the first iteration of Brajard et al. 2020' loop.

28. l.236-242: What you intend to do is not clear enough to me. Please be more specific. For instance, give the corresponding algorithm. What is the model used by the LETKF supposed to be?

29. p.15: A state space dimension of $m = 8$ is small while the ensemble size $N_e = 20$ is quite large for such state space size. That makes localisation totally unnecessary if not useless. This is really problematic as a demonstration of the method.

30. l.253: "F term in" ⟶ "The F term in"

31. l.253-254: "F term in the model was changed to represent the model bias.": This is only one degree of freedom. Any good data assimilation method can handle this without resorting to machine learning, see for instance Bocquet and Sakov (2013). This undermines your demonstration to some extent.

32. l.262, Eq.(15): the work/concept of "localisation" has not been mentioned once. Has it?

33. l.267: "The configuration of RC used in this study was similar" ⟶ "The configuration of RC used in this study is similar"

34. l.267, Eq.(16): The words "odd" and 'even" need not be italicised.

35. From line 271 till the end of the numerical section, please use the present tense, not the past tense.

36. page 16: You could also use an (L)ETKF that accounts for model error, for instance that of Raanes et al. (2015). All NWP data assimilation methods have model error correction steps.

37. l.295: What you refer to is the "forecast lead time".

38. l.312: It seems to me that you cannot refer to figures of the supplement material. Your paper has to be self-contained. Please incorporate the figures or remove the paragraph. You may want to discuss this issue with the Editor.

39. l.324: "under imperfect models": do you mean "perfect model"? This is very confusing because you just discussed perfect model experiments and will do so again in the next sentence of the same paragraph (figure 5). It is only in the next paragraph that you report experiments with imperfect model. Please improve the text.

40. l.334-335: To which values do you change F?

41. Figure 2: Where are the tags (a), (b), ..., (e) in the panels?

42. Figure 4: You could use a log scale for the y-axis (RMSE) to see what happens for the shorter forecast lead times.

43. l.336-338 and Figure 6: The comparison is visually difficult. I suggest to merge Figure 6a and 6b which should facilitate the comparison.

44. l. 351: You cannot refer to figures of the supplement material. Your paper has to be self-contained. Please incorporate the figures or remove the paragraph.

45. l.388-390: No, in their method one can limit the number of iterations, to 1 for instance, just as you do. These authors also have a sequel to this paper, where they use a physical imperfect model as a first iteration, just as you do and this is potentially applicable to high-dimensions (Brajard et al., 2020b). By the way, you are using m=8 while they are using m=40, so they are a little closer to high-dimension than you are.

46. l.390: "contrary" ⟶ "By contrast"

47. l.406: "The advantage of our proposed method is that we allow both models and observation networks to be imperfect.": So do a lot of other methods that use other ML technique than RC (Bocquet et al., 2019; Brajard et al., 2020a; Bocquet et al., 2020) for instance.

48. l.413-416: When reading the abstract of your paper, I honestly thought that you would use the local RC, since you were claiming to use the LETKF. Doing so would have make your paper much stronger and more consistent (using a local RC with a local ETKF). Why did you not try it?

49. l.421-422: "These results imply that our proposed method can be applicable to various realistic problems.": Testing the method on a 8-dimensional problem with a global RC does not make it applicable to various realistic problems! Please mitigate this too bold statement.

50. l.431-433: "Our new method is robust to the imperfectness of both models and observations so that it is feasible to apply it to the real NWP problem.": I really don't believe you can make such statement, from an 8-dimensional L96 model. Please remove that statement, which could be shocking to many colleagues working in numerical weather prediction and data assimilation.

**References**

Bocquet, M., Brajard, J., Carrassi, A., Bertino, L., 2019. Data assimilation as a learning tool to infer ordinary differential equation representations of dynamical models. Nonlin. Processes Geophys. 26, 143–162. doi:`10.5194/npg-26-143-2019`.

Bocquet, M., Farchi, A., Malartic, Q., 2020. Online learning of both state and dynamics using ensemble kalman filters. Foundations of Data Science 0, 00–00. URL: https://arxiv.org/pdf/2006.03859.pdf. accepted for publication.

Bocquet, M., Sakov, P., 2013. Joint state and parameter estimation with an iterative ensemble Kalman smoother. Nonlin. Processes Geophys. 20, 803–818. doi:`10.5194/npg-20-803-2013`.

Brajard, J., Carrassi, A., Bocquet, M., Bertino, L., 2020a. Combining data assimilation and machine learning to emulate a dynamical model from sparse and noisy observations: a case study with the Lorenz 96 model. J. Comput. Sci. 44, 101171. doi:`10.1016/j.jocs.2020.101171`.

Brajard, J., Carrassi, A., Bocquet, M., Bertino, L., 2020b. Combining data assimilation and machine learning to infer unresolved scale parametrisation. Philosophical Transactions A 0, 0. URL: https://arxiv.org/pdf/2009.04318.pdf. submitted.

Lorenz, E.N., Emanuel, K.A., 1998. Optimal sites for supplementary weather observations: simulation with a small model. J. Atmos. Sci. 55, 399–414. doi:`10.1175/1520-0469(1998)055<0399:OSFSWO>2.0.CO;2`.

Raanes, P.N., Carrassi, A., Bertino, L., 2015. Extending the square root method to account for additive forecast noise in ensemble methods. Mon. Wea. Rev. 143, 3857–38730. doi:`10.1175/MWR-D-14-00375.1`.

---

## Author Comment (AC1) · 13 Nov 2020

Response letter of gmd-2020-211

Dear Editor,

Please find the revised version of our manuscript "Combining Ensemble Kalman Filter and Reservoir Computing to predict spatio-temporal chaotic systems from imperfect observations and models", which we would like to resubmit for publication in *Geoscientific Model Development.*

Comments made by the reviewers were highly insightful. They allowed us to greatly improve the quality of the manuscript. We described the responses to the comments.

Each comment made by the reviewers is written in *italic* font. We numbered each comment as (n.m) in which n is the reviewer number and m is the comment number. In the revised manuscript, changes are highlighted in yellow in the single-column and double-spaced paper.

We trust that the revisions and responses are sufficient for this manuscript to be published in *Geoscientific Model Development*.

Sincerely,
Futo Tomizawa, Yohei Sawada

**Responses to the comments of Referee #1**

*(1.1) Section 2.4 "Combination of RC and LETKF" is the scientific innovation that is at the core of the paper, however the authors only explore the idea in a purely textual form. Some equations comparing the three methods (RC-Obs, RC-Anl, LETKF) or perhaps a diagram would be immensely beneficial to the average reader, as that will be more eye-catching, and would help explain the papers innovation.*

$\rightarrow$ The prediction schemes of RC-Obs, RC-Anl, and LETKF are compared in Table 2 in the original version of manuscript. However, there is the room to improve the description of the idea of RC-Anl. We have added some equations to explain the concept of each prediction scheme, LETKF-Ext, RC-Obs, and RC-Anl.

Lines 192-196: Predictions are made by the model alone, using the latest analysis state variables as the initial condition:

$$x^f_{K+1} = \widetilde{\mathcal{M}}(\overline{x^a_K}), \quad x^f_{K+2} = \widetilde{\mathcal{M}}(x^f_{K+1}), \quad \dots$$

where $x^f_k$ is the prediction variables at time $k$, $\widetilde{\mathcal{M}}$ is the prediction model (an imperfect L96 model), and $\overline{x^a_K}$ is the mean of the analysis ensemble at the initial time of the prediction.

Lines 243-247: At this point, RC can now be used as the surrogate model that mimics the state dynamics:

$$x^f_{k+1} = \widetilde{\mathcal{M}}_{RC}\left(x^f_k, \{x^{train}_k\}_{1 \le k \le K}\right) \tag{16}$$

where $x^f_k$ is the prediction variables at time $k$, $\widetilde{\mathcal{M}}_{RC}$ is the dynamics of RC (equations (12) and (13)) and $\{x^{train}_k\}_{1 \le k \le K} = \{x^{train}_1, x^{train}_2, \dots, x^{train}_K\}$ is the time series of training data.

Lines 251-254: Prediction time series here can be expressed using equation (16) as follows:

$$x^f_{K+1} = \widetilde{\mathcal{M}}_{RC}\left(y^O_K, \{y^O_k\}_{1 \le k \le K}\right), \quad x^f_{K+2} = \widetilde{\mathcal{M}}_{RC}\left(x^f_{K+1}, \{y^O_k\}_{1 \le k \le K}\right), \dots \tag{17}$$

where $\{y^O_k\}_{1 \le k \le K} = \{y^O_1, y^O_2, \dots, y^O_K\}$ is the observation time series and $y^O_K$ is the observation at the initial time of the prediction.

Lines 292-304: Suppose we have sparse and noisy observations for the training data. If we take observations as inputs and analysis variables as outputs, LETKF can be considered as an operator to estimate the full state variables from the sparse observations:

$$\{\overline{x^a_k}\}_{1 \le k \le K} = \{\mathcal{D}(y^O_k)\}_{1 \le k \le K} \tag{22}$$

where $\{\overline{x^a_k}\}_{1 \le k \le K} = \{x^a_1, x^a_2, \dots, x^a_K\}$ is the full-state variables (time series of the LETKF analysis ensemble mean), $y^O_k$ is the observation, and $\mathcal{D}: \mathbb{R}^n \rightarrow \mathbb{R}^m$ represents the state estimation operator, which is realized by LETKF in this study. Then, RC is trained by using $\{x^a_k\}_{1 \le k \le K}$ as the training data set. In this way, RC can mimic the dynamics of analysis time

series computed by forecast-analysis cycle of LETKF. Prediction can be generated by using the analysis variables at current time step ($x_K^a$) as the initial value. Since RC is trained with LETKF analysis variables, we call this method "RC-Anl". By using the notation of equation (16), the prediction of RC-Anl can be expressed as follows:

$$x_{K+1}^f = \widetilde{\mathcal{M}}_{RC}(x_K^a, \{x_k^a\}_{1 \leq k \leq K}), \qquad x_{K+2}^f = \widetilde{\mathcal{M}}_{RC}(x_{K+1}^f, \{x_k^a\}_{1 \leq k \leq K}), \dots \qquad (23)$$

where $\{x_k^a\}_{1 \leq k \leq K} = \{x_1^a, x_2^a, \dots, x_K^a\}$ is the time series of the LETKF analysis variables.

In addition, we have added a flow chart that explains the procedures of LETKF-Ext, RC-Obs, RC-Anl.

Lines 324-325: The schematics of the LETKF-Ext, RC-Obs, and RC-Anl are shown in the Figure 3. Initial values and model dynamics used in each method are compared in Table 1.

[Figure]

**Figure 3**. The algorithm flow of LETKF-Ext, RC-Anl, and RC-Obs. Solid and dotted lines show the flow of variables and models (either process-based or data-driven surrogate), respectively.

*(1.2) Section 3 "Experimental Design" mention the parameters used to construct the RC network, however motivations for the choices are lacking, and would help potential follow-up works, including those by practitioners to intuit logical choices.*

→ The parameter values used in this study is chosen based on the settings in previous works such as Vlachas et al., 2020 (with the adoption of parallelized Reservoir Computing, we changed the reference from Chattopadhyay et al., 2019 to Vlachas et al., 2020), and we had confirmed that these parameters give the good accuracy by performing some sensitivity tests. Through these sensitivity tests, we found that most of the parameters do not substantially affect the accuracy of the prediction. One effective parameter is the spectral radius of the reservoir network ($\rho$). Jiang and Lai (2019) reported that $\rho$ needs to be tuned to obtain good prediction accuracy. We also verified this fact and found the proper

value for $\rho$ by the sensitivity tests. This point was indeed unclear in the original version of the paper and has been clarified in the revised version of the paper.

> Lines 342-347: Jiang and Lai (2019) revealed that the performance of RC is sensitive to $\rho$ and it needs to be tuned. We identified the proper value of $\rho$ by sensitivity studies. Other parameters do not substantially affect the prediction accuracy, and we selected them based on the settings in previous works such as Vlachas et al. (2020). The nonlinear transformation function for the output layer in equation (13) is the same as Chattopadhyay et al. (2019), which is represented as follows:

*(1.3) Equation (15) represents Gaussian kernel localization, however this is not explicitly mentioned, which seems very strange, as a reader familiar with data assimilation literature would be more familiar with that framing of the method. Additionally, such applications are typically done on the inverse of the matrix R (which is also explicitly taken to be diagonal), through a Woodbury matrix identity application, though such details are glossed over in the text.*

→ Since the actual procedure of localization process in LETKF was not clearly mentioned in the original version of the paper, we have added the explanations in the revised version of the paper. The general description of the smooth localization is discussed in section 2, following the comment (3.33).

> Lines 329-335: As the localization process, the observation point within 10 indices are chosen to be assimilated for every grid point. The "smooth localization" is also performed on observation covariance $R$. Since we assume that each observation error is independent and thus $R$ is diagonal, the localization procedure can be done just by dividing each diagonal elements of observation covariance $R$ by the value *w* calculated as follows:
>
> $$w(r) = \exp\left(-\frac{r^2}{18}\right) \tag{24}$$
>
> where $r$ is the distance between each observation point and each analyzed point. For every grid point, the observation point with $w(r) \geq 0.0001$ are chosen to be assimilated.

However, to our best knowledge, "Gaussian Kernel Localization" is not a common term in the data assimilation community of geoscience, meteorology, or hydrology. We did not use this word in the revised version of the manuscript. It would be very helpful if the reviewer will give us some references that use the term "Gaussian Kernel Localization" in the next iteration of the peer-review process.

*(1.4) The strangest thing is the choice of $m = 8$ for the experimental design. The standard L96 with $m = 40$ variables is known to be very chaotic and have 13 positive Lyapunov exponents. In that regime, a lot is known about the system. However as far as the reviewer can tell, the case of $m = 8$ as to be weakly chaotic by necessity. It would be of benefit to the casual reader (and the reviewer) to state some facts about this regime of the system.*

→ As the reviewer pointed out, $m = 8$ is not commonly used in the previous studies. We conducted new experiments with the 40-dimensional L96 system with parallelized reservoir computing which is more suitable for the prediction of higher dimensional systems. We have added the description of the 40-dimensional L96 system introduced by Lorenz & Emanuel (1998).

>Lines 105-110: It is known that the model with $m = 40$ and $F = 8$ shows chaotic dynamics with 13 positive Lyapunov exponents (Lorenz & Emanuel, 1998), and this setting is commonly used in the previous studies (e.g. Kotsuki et al., 2017; Miyoshi, 2005; Penny, 2014; Raboudi et al., 2018). The time width $\Delta t = 0.2$ corresponds to one day in real atmospheric motion from the view of dissipative decay time (Lorenz & Emanuel, 1998).

In the revised version of the paper, all the sentences that refer to the dimensions of state space have been modified, and all results have been updated to the 40-dimension L96 system. Figures 4, 5, 7-10 (Figures 2-7 in the original paper) have been replaced in the revised version of the paper the revised figures are shown below. It should be noted that the primary findings of the original paper do not change when we changed the dimension of the L96 system.

[Figure]

**Figure 4.** The Hovmöller diagram of (a) Nature Run, (b) A prediction of RC-Obs, (c) difference of (a) and (b). Horizontal axis shows the timesteps and vertical axis shows the nodal number. Value at each timestep and node is represented by the color.

[Figure]

**Figure 5**. The *mRMSE* time series of the predictions of RC-Obs with perfect observation. Horizontal axis shows the timestep and vertical shows the value of *mRMSE*.

[Figure]

**Figure 7**. The *mRMSE* time series of the predictions of (a)LETKF-Ext and (b)RC-Obs with spatially sparse observation. Each color corresponds to the number of the observation points.

[Figure]

**Figure 8**. The same as figure4, for the RC-Anl prediction.

[Figure]

**Figure 9**. The *mRMSE* time series of the predictions of (a)LETKF-Ext and (b)RC-Anl with biased model. Each color corresponds to each value of *F* term.

[Figure]

**Figure 10**. The *mRMSE(80)* of the predictions of LETKF-Ext(red) and RC-Anl(blue) for each model bias. Horizontal axis shows the value of the force parameter of equation (1) (8 is the true value) and vertical axis shows the value of *mRMSE*.

*(1.5) The reviewer wonders if for such a small system it might have been of benefit to examine the more rigorously studied 3-variable Lorenz system, for which a lot is known, in terms of reservoir computing, data assimilation, and general nature of the system. The same type of sparsity experiments could be performed with such a system.*

→ We believe that the 3-variable Lorenz system is too small for our study. Figures 4 and 5 in the original paper shows the result of predictions in which we reduce the observable points one by one up to a half of full state. It is difficult to perform this sensitivity study with only 3 state variables. Since our future prospect is to extend the proposed method to be applicable to much higher-dimensional systems such as the NWP problem, we have decided to increase the dimension rather than decrease it

to the 3-variable Lorenz system. As mentioned in our response to the comment (1.4), we conducted new experiments with 40-dimensional L96 system in the revised version of the paper. See our response to (1.4). We have decided not to change the paper responding to this comment.

*(1.6) Section 4 "Results" contains a statement to the effect of "if 'e' is set to zero then the LETKF does not work". This is technically true, but very misleading. Take the Kalman filter formula with perfect and full observations $x^a = x^f - (x^f - y^o) = y^o$. Under the regime of perfect observation ( $e = 0$, $H = I$ ), and Kalman filter would replace the analysis by the observations, and collapse the distribution of the uncertainty about the analysis. This means that in this regime, the LETKF would replace each ensemble member by the observations, even if a particular implementation would fail. It would make sense to—instead of using the LETKF—to simply look at the predictive regime of the RC-Obs. The $e = 0.01$ is entirely unnecessary.*

$\rightarrow$ We agree with this reviewer's comment and we have deleted Figure 2 of the original version of the paper. In many previous studies that applied machine learning methods to predict chaotic systems, observations of full state space with no noise are often assumed so that we performed RC-Obs with these perfect observations. Since we compared RC and LETKF with many different settings, we guessed that the reader might want us to compare them under these perfect observations. Therefore, we performed the LETKF with almost perfect observations (e=0.01) in the original version of the paper although it is essentially not meaningful as the reviewer pointed out.

For readers of Geoscientific Model Development, who are not necessarily familiar with data assimilation methods, we have added some explanations which we discussed above.

> Line 380-384: Figure 4 shows the Hovmöller diagram of a prediction of RC-Obs and Nature Run. Figure 4 also shows the difference between prediction and Nature Run, as well as the actual prediction results so that we can see how long we can keep the prediction accurate. RC is trained with perfect observation ($e = 0$ at all grid point). Figure 4 shows that RC-Obs predicts accurately within approximately 200 timesteps.
>
> Line 386-391: Figure 5 shows the time variation of the *mRMSE* (see equation (26)) of RC-Obs with perfect observation. It also shows that RC-Obs can predict with good accuracy for approximately 200 timesteps. It should be noted that LETKF (as well as other data assimilation methods) is just the model's forecast with the initial conditions identical to Nature Run when all state variables can be perfectly observed, and thus the prediction accuracy of LETKF-Ext will be perfect if we have no model bias. LETKF-Ext is much superior to RC-Obs under this regime (not shown).

*(1.7) The choice of which indices are observed in such experiments is not given as far as the reviewer can tell.*

→Observed grid points are chosen to maintain the uniformity of the observation. This point was indeed unclear in the original version of the paper. We have added a table showing the observed indices..

> Line 402-403: ==Observation is reduced as uniformly as possible. The observation network in each experiment is shown in Table 3.==

**Table 3.** The indices of observed grid points.

| # Observed | Grid point index | | | | | | | | | | | | | | | | | | | | | | | | | | | | | | | | | | | | | | | |
|---|---|---|---|---|---|---|---|---|---|---|---|---|---|---|---|---|---|---|---|---|---|---|---|---|---|---|---|---|---|---|---|---|---|---|---|---|---|---|---|---|
| | 1 | 2 | 3 | 4 | 5 | 6 | 7 | 8 | 9 | 10 | 11 | 12 | 13 | 14 | 15 | 16 | 17 | 18 | 19 | 20 | 21 | 22 | 23 | 24 | 25 | 26 | 27 | 28 | 29 | 30 | 31 | 32 | 33 | 34 | 35 | 36 | 37 | 38 | 39 | 40 |
| 40 | • | • | • | • | • | • | • | • | • | • | • | • | • | • | • | • | • | • | • | • | • | • | • | • | • | • | • | • | • | • | • | • | • | • | • | • | • | • | • | • |
| 38 | • | • | • | • | • | • | • | • | • | • | • | • | • | • | • | • | • | • | • |  | • | • | • | • | • | • | • | • | • | • | • | • | • | • | • | • | • | • | • |  |
| 36 | • | • | • | • | • | • | • | • | • |  | • | • | • | • | • | • | • | • | • |  | • | • | • | • | • | • | • | • | • |  | • | • | • | • | • | • | • | • | • |  |
| 30 | • | • | • |  | • | • | • |  | • | • | • |  | • | • | • |  | • | • | • |  | • | • | • |  | • | • | • |  | • | • | • |  | • | • | • |  | • | • | • |  |
| 20 | • |  | • |  | • |  | • |  | • |  | • |  | • |  | • |  | • |  | • |  | • |  | • |  | • |  | • |  | • |  | • |  | • |  | • |  | • |  | • |  |

*(1.8) In section 5 "Discussion" the authors mention the generalization of their methodology to other dynamical systems. The authors focus on the generalization to "higher dimensional systems" as their main worry. The reviewer believes that another more substantial worry is that Lorenz '96 is known to be ergodic, while most systems used for NWP are possibly not, (the authors mention the KS system, but that is also possibly ergodic for well-posed initial conditions). It would be of benefit to mention other possible shortcoming on testing only on the Lorenz '96 system.*

→ We agree that the main concern of testing only on the L96 is that this model is ergodic. However, it is known that the 40-dimensional L96 shows strong chaos and large Lyapunov exponents, and this model has been used as a good initial testbed toward the application to the large and non-ergodic NWP problems. In the revised version of the manuscript, the scalabilities in terms of dimensionality and ergodicity are discussed separately, and main shortcomings on testing only on the L96 system are explicitly mentioned.

> Lines 505-508: ==Although we tested our method only on 40-dimensional Lorenz 96 system, Pathak, Hunt et al. (2018) indicated that parallelized RC can be extended to predict the dynamics of substantially high dimensional chaos such as 200-dimensional Kuramoto-Sivashinski equation with small computational costs. It implies that the findings of this study can also be applied to higher dimensional systems.==

> Lines 515-518: ==However, since the Lorenz 96 model (and other conceptual models such as Kuramoto-Sivashinski equation) is ergodic, it is unclear that our method can be applied to real NWP problems directly, which are possibly non-ergodic. Although our proposed method has a potential to extend to larger and more complex problems, further studies are needed.==

**Responses to the comments of Referee #2**

*The summary of existing work, novel directions, and discussion of findings is strong. Some further questions: Does learning the LETFK analysis with a different method (e.g. vanilla feedforward network) give the same result? In other words, is there anything interesting about reservoir computing in particular in this context? Does learning the analysis from some other method work in the same way?*

→ We thank the reviewer for highly evaluating the paper. We believe that our method has flexibility in choosing machine learning methods, since our method does not depend on a specific framework. If other machine learning methods perform as good as the RC implementation of our study, replacing RC with them would give similar results. However, the training of RC is computationally cheaper than the other neural networks, which is the significant advantage of our strategy toward the real-world applications. Although Vlachas et al. (2020) pointed out that RC is more vulnerable to the sparse observation networks than the other neural network methods, we overcome this issue by combining it with data assimilation. Therefore, we believe that the use of RC is particularly interesting in this context although we can directly carry out to learn from the LETKF analysis using other neural networks. This point has been clarified in the revised version of the paper.

> Lines 476-486: ==Note also that the computational cost to train RC is much cheaper than the other neural networks. Since the framework of our method does not depend on a specific machine learning framework, we believe that we can flexibly choose other machine learning methods such as RNN, LSTM, ANN, etc. Previous studies such as Chattopadhyay et al., (2019) or Vlachas et al., (2020) revealed that these methods show competitive performances compared to RC in predicting spatio-temporal chaos. Using them instead of RC in our method would probably give similar results. However, the advantage of RC is its cheap training procedure. RC does not need to perform an expensive back-propagation method for training, unlike other neural networks (Lu et al., 2017; Chattopadhyay et al., 2019). Therefore, RC is considered as a promising tool for predicting spatio-temporal chaos. Although our method has flexibility in the choice of machine learning methods, we consider that the good performance with RC is important in this research context.==

We can say similar things if we use different data assimilation methods. We consider that the LETKF can effectively estimate state variables using imperfect models and observations, although it cannot modify the model bias itself when the sources of bias are unknown. To our best knowledge, they are general features of data assimilation methods. Thus, we believe that replacing LETKF with other data assimilation methods will give us similar results as in this study. Since we believe that this feature of sequential data assimilation is well known in the data assimilation community, we have not included this point explicitly for brevity.

**Responses to the comments of Referee #3**

*1. Major issues*

*In spite of the obvious care taken to write this manuscript, it has quite a few issues, mainly but not only related to the presentation of the theory and to the choices made for the numerical experiments:*

*(3.1) I don't see the point in presenting the Kalman filter. Why not go straight to the introduction of the (L)ETKF? The introduction of the Kalman filter is really unnecessary.*

→ In the original version of the paper, we thought that some explanations of very basics of Kalman Filter is necessary for readers who were not familiar to data assimilation. We agree with this reviewer's comment that our explanation was circuitous. We have deleted the introduction of Kalman Filter and explained LETKF directly in the revised version of the manuscript.

Linear approximation of the observation operator is also explained only in the Kalman Filter section. We have added some descriptions to introduce this assumption.

> Lines 151-153: Although $\mathcal{H}$ can be either linear or nonlinear, we assume it to be linear in this study and expressed as a $h \times m$ matrix $\boldsymbol{H}$ (the treatment of the nonlinear case is discussed in Hunt et al., 2007).

The descriptions of forecast step in LETKF is modified to explain without using the equation of Kalman Filter.

> Lines 155-162-: LETKF uses an ensemble of state variables to estimate the evolution of $\overline{\boldsymbol{x}_k^f}$ and $\boldsymbol{P}_k^f$. The time evolution of each ensemble members is as follows:
>
> $$\boldsymbol{x}_k^{f,(i)} = \mathcal{M}\left(\boldsymbol{x}_{k-1}^{a,(i)}\right) \qquad (5)$$
>
> where $\boldsymbol{x}_k^{f,(i)}$ is the $i$th ensemble member of forecast value at time $k$. Then the mean and covariance of state variables can be expressed as follows:
>
> $$\overline{\boldsymbol{x}_k^f} \approx \frac{1}{N_e}\sum_{i=1}^{N_e} \boldsymbol{x}_k^{f,(i)}, \qquad \boldsymbol{P}_k^f = \frac{1}{N_e-1}X_k^f\left(X_k^f\right)^T \qquad (6)$$
>
> where $N_e$ is the number of ensemble members and $X_k^f$ is the matrix whose $i$th column is the deviation of the $i$th ensemble member from the ensemble mean.

The description of the analysis step of LETKF is also modified in the form that does not depend on the explanation of Kalman Filter. However, the modification of this part is more relevant to the comment (3.3). See also the corresponding response.

*(3.2) The introduction of the LETKF can be perfected (see a few points below).*

→ We have modified the introduction of LETKF following the reviewer's indications. See the response to (3.1) and (1.3), which is shown below.
* * *
*(1.3) Equation (15) represents Gaussian kernel localization, however this is not explicitly mentioned, which seems very strange, as a reader familiar with data assimilation literature would be more familiar with that framing of the method. Additionally, such applications are typically done on the inverse of the matrix R (which is also explicitly taken to be diagonal), through a Woodbury matrix identity application, though such details are glossed over in the text.*

→ Since the actual procedure of localization process in LETKF was not clearly mentioned in the original version of the paper, we have added explanations in the revised version of the paper. The general description of the smooth localization is discussed in section 2, following the comment (3.33).

Lines 329-335: As the localization process, the observation point within 10 indices are chosen to be assimilated for every grid point. The "smooth localization" is also performed on observation covariance $R$ Since we assume that each observation error is independent and thus $R$ is diagonal, the localization procedure can be done just by dividing each diagonal elements of observation covariance $R$ by the value $w$ calculated as follows:

$$w(r) = \exp\left(-\frac{r^2}{18}\right) \tag{24}$$

where $r$ is the distance between each observation point and each analyzed point. For every grid point, the observation point with $w(r) > 0.0001$ are chosen to be assimilated.

However, to our best knowledge, "Gaussian Kernel Localization" is not a common term in the data assimilation community of geoscience, meteorology, or hydrology. We did not use this word in the revised version of the manuscript. It would be very helpful if the reviewer will give us some references that use the term "Gaussian Kernel Localization" in the next iteration of the peer-review process.
* * *
*(3.3) Localisation is never explicitly mentioned, which is at serious odds with the use of the LETKF.*
→ The explanation of localization was indeed unclear in the original version of the paper. We have added the detailed explanation of localization in section 2.

Lines 164-170: In the analysis step, LETKF assimilates only the observations close to each grid point. Therefore, the assimilated observations are different at different grid points and the analysis variables of each grid points are computed separately.

For each grid points, observations to be assimilated are chosen. The rows or elements of $\boldsymbol{y}^o$, $\boldsymbol{H}$, and $\boldsymbol{R}$ corresponding to non-assimilated observations should be removed as the localization procedure. "Smooth localization" can also be performed by multiplying some factors to each element of $\boldsymbol{R}$ based on the distance between target grid point and observation points.

The actual procedure of localization used in our experiment is explained in section 3 more explicitly in the revised version of the manuscript. See also our response to (3.2).

*(3.4) The description of the main algorithm is not very precise. I believe that it could be significantly improved.*

→ The description of RC-Anl in the original paper was just a textual form and the explanation was not sufficient. We have modified the description of the algorithm in the revised version of the paper, along with a new figure that compares the implementations of LETKF-Ext, RC-Anl, and RC-Obs. See our response to (1.1) shown below.
* * *
*(1.1) Section 2.4 "Combination of RC and LETKF" is the scientific innovation that is at the core of the paper, however the authors only explore the idea in a purely textual form. Some equations comparing the three methods (RC-Obs, RC-Anl, LETKF) or perhaps a diagram would be immensely beneficial to the average reader, as that will be more eye-catching, and would help explain the papers innovation.*

→ The prediction schemes of RC-Obs, RC-Anl, and LETKF are compared in Table 2 in the original version of manuscript. However, there is the room to improve the description of the idea of RC-Anl. We have added some equations explaining the concept of each prediction scheme, LETKF-Ext, RC-Obs, and RC-Anl.

Lines 192-196: Prediction are made by the model alone, using the latest analysis state variables as the initial condition:

$$x^f_{K+1} = \widetilde{\mathcal{M}}(\overline{x^a_K}), \ \ x^f_{K+2} = \widetilde{\mathcal{M}}(x^f_{K+1}), \ \ ...$$

where $x^f_k$ is the prediction variables at time $k$, $\widetilde{\mathcal{M}}$ is the prediction model (an imperfect L96 model), and $\overline{x^a_K}$ is the mean of the analysis ensemble at the initial time of the prediction.

Lines 243-247: At this point, RC can now be used as the surrogate model that mimics the state dynamics:

$$x^f_{k+1} = \widetilde{\mathcal{M}}_{RC}\left(x^f_k, \{x^{train}_k\}_{1 \le k \le K}\right) \tag{16}$$

where $\boldsymbol{x}_k^f$ is the prediction variables at time $k$, $\widetilde{\mathcal{M}}_{RC}$ is the dynamics of RC (equations (12) and (13)) and $\left\{\boldsymbol{x}_k^{train}\right\}_{1\le k\le K} = \left\{\boldsymbol{x}_1^{train}, \boldsymbol{x}_2^{train}, \dots, \boldsymbol{x}_K^{train}\right\}$ is the time series of training data.

Lines 251-254: Prediction time series here can be expressed using equation (16) as follows:

$$\boldsymbol{x}_{K+1}^f = \widetilde{\mathcal{M}}_{RC}\left(\boldsymbol{y}_K^O, \{\boldsymbol{y}_k^O\}_{1\le k\le K}\right), \quad \boldsymbol{x}_{K+2}^f = \widetilde{\mathcal{M}}_{RC}\left(\boldsymbol{x}_{K+1}^f, \{\boldsymbol{y}_k^O\}_{1\le k\le K}\right), \dots$$

where $\{\boldsymbol{y}_k^O\}_{1\le k\le K} = \{\boldsymbol{y}_1^O, \boldsymbol{y}_2^O, \dots, \boldsymbol{y}_K^O\}$ is the observation time series and $\boldsymbol{y}_K^O$ is the observation at the initial time of the prediction.

Lines 292-304: Suppose we have sparse and noisy observations for the training data. If we take observations as inputs and analysis variables as outputs, LETKF can be considered as an operator to estimate the full state variables from the sparse observations:

$$\{\overline{\boldsymbol{x}_k^a}\}_{1\le k\le K} = \{\mathcal{D}(\boldsymbol{y}_k^O)\}_{1\le k\le K} \tag{22}$$

where $\{\overline{\boldsymbol{x}_k^a}\}_{1\le k\le K} = \{\boldsymbol{x}_1^a, \boldsymbol{x}_2^a, \dots, \boldsymbol{x}_K^a\}$ is the full-state variables (time series of the LETKF analysis ensemble mean), $\boldsymbol{y}_k^O$ is the observation, and $\mathcal{D}: \mathbb{R}^n \to \mathbb{R}^m$ represents the state estimation operator, which is realized by LETKF in this study. Then, RC is trained by using $\{\boldsymbol{x}_k^a\}_{1\le k\le K}$ as the training data set. In this way, RC can mimic the dynamics of analysis time series computed by forecast-analysis cycle of LETKF. Prediction can be generated by using the analysis variables at current time step $(\boldsymbol{x}_K^a)$ as the initial value. Since RC is trained with LETKF analysis variables, we call this method "RC-Anl". By using the notation of equation (16), the prediction of RC-Anl can be expressed as follows:

$$\boldsymbol{x}_{K+1}^f = \widetilde{\mathcal{M}}_{RC}(\boldsymbol{x}_K^a, \{\boldsymbol{x}_k^a\}_{1\le k\le K}), \qquad \boldsymbol{x}_{K+2}^f = \widetilde{\mathcal{M}}_{RC}(\boldsymbol{x}_{K+1}^f, \{\boldsymbol{x}_k^a\}_{1\le k\le K}), \dots$$

where $\{\boldsymbol{x}_k^a\}_{1\le k\le K} = \{\boldsymbol{x}_1^a, \boldsymbol{x}_2^a, \dots, \boldsymbol{x}_K^a\}$ is the time series of the LETKF analysis variables.

In addition, we have added a flow chart that explains the procedures of LETKF-Ext, RC-Obs, RC-Anl. The Table 1 in the original version of the manuscript is deleted since the Figure 3 contains the information that was explained in the table.

Lines 324-325: The schematics of the LETKF-Ext, RC-Obs, and RC-Anl are shown in the Figure 3. Initial values and model dynamics used in each method are compared in Table 1.

[Figure]

==Figure 3==. ==The algorithm flow of LETKF-Ext, RC-Anl, and RC-Obs. Solid and dotted lines show the flow of variables and models (either process-based or data-driven surrogate), respectively.==
* * *
*(3.5) The statements about the applicability of the method in the discussion and the conclusion are much too strong, and should either be retracted or significantly mitigated.*

→ We agree that our statements on the scalability of our method in the sections 5 and 6 are too strong considering the numerical experiments conducted in our study. We have mitigated or modified the expressions in these sections following the suggestions by the reviewer. See the following responses to (3.54), (3.56), and (3.57).

*(3.6) The numerical experiment choices show that localisation is totally unnecessary here which significantly undermines the manuscript. Using the ETKF would have yielded results at least as good. In particular any claim of applicability to higher dimensional models the authors make in the discussion and conclusion is seriously undermined by this issue (see specific points below). In this vein, it is rather disappointing that the authors did not apply the local RC (together with the LETKF) which would have been a significant added value to the paper. This is what I initially expected from the title and abstract of the paper.*

→ We agree that the localization is unnecessary for the 8-dimensional L96 system. We have conducted new experiments with commonly used the 40-dimensional L96 system. As the target system was enlarged, we have adopted the parallelized reservoir computing (which we believe the reviewer indicated by the term "local RC") as the RC architecture.

The Abstract is modified to clearly show that we used parallelized reservoir computing.

> Lines 19-20: ==In order to increase the scalability to larger systems, we applied parallelized RC framework.==

The original papers from Pathak, Lu et al., (2018) that introduces the parallelized reservoir approach is explicitly cited in the introduction section.

Lines 59-61: Pathak, Lu et al. (2018) succeeded in using a parallelized RC to predict each segment of the state space locally, which enhanced the scalability of RC to much higher dimensional systems.

The introduction of parallelized Reservoir Computing is shown right after the description of serial RC, along with a conceptual diagram.

Lines 256-276 : **2.3.1    Parallelized Reservoir Computing**

In general, the required reservoir size $D_r$ for accurate prediction increases as the dimension of the state space $m$ increases. Since the RC framework needs to keep adjacency matrix $\boldsymbol{A}$ on the memory, and to perform inverse matrix calculation of $D_r \times D_r$ matrix (equation (15)), too large reservoir size leads to unfeasible computational cost. Pathak, Hunt et al. (2018) proposed a solution to this issue, which is called the parallelized reservoir approach.

In this approach, the state space is divided into $g$ groups, all of which contains $q = m/g$ state variables:

$$\boldsymbol{g}_k^{(i)} = \left(u_{k,(i-1)\times q+1}, u_{k,(i-1)\times q+2}, \dots, u_{k,i\times q}\right)^T, i = 1, 2, \dots, g \tag{18}$$

where $\boldsymbol{g}_k^{(i)}$ is the $i$th group at time $k$, $u_{k,j}$ is the $j$th state variable at time $k$. Each group is predicted by different reservoir placed in parallel. $i$th reservoir accepts the state variables of $i$th group as well as adjacent $l$ grids, which can be expressed as follows:

$$\boldsymbol{h}_k^{(i)} = \left(u_{k,(i-1)\times q+1-l}, u_{k,(i-1)\times q+2-l}, \dots, u_{k,i\times q+l}\right)^T \tag{19}$$

where $\boldsymbol{h}_k^{(i)}$ is the input vector for $i$th reservoir at time $k$. The dynamics of each reservoir can be expressed as follows according to equation (12):

$$\boldsymbol{r}_{k+1}^{(i)} = \tanh\left[\boldsymbol{A}^{(i)}\boldsymbol{r}_k^{(i)} + \boldsymbol{W}_{in}^{(i)}\boldsymbol{h}_k^{(i)}\right] \tag{20}$$

where $\boldsymbol{r}_k^{(i)}$, $\boldsymbol{A}^{(i)}$, $\boldsymbol{W}_{in}^{(i)}$ and $\boldsymbol{W}_{out}^{(i)}$ are the reservoir state vector, adjacency matrix, input matrix, and output matrix for $i$th reservoir. Each reservoir is trained independently using equation (13) so that:

$$\boldsymbol{g}_k^{(i)} = \boldsymbol{W}_{out}^{(i)}\boldsymbol{f}\left(\boldsymbol{r}_k^{(i)}\right) \tag{21}$$

where $\boldsymbol{W}_{out}^{(i)}$ is the output matrix in the $i$th reservoir. The prediction scheme of parallelized RC is summarized in Figure 2.

[Figure]

**Figure 2**. The conceptual diagram of parallelized reservoir computing architecture. The state space is separated into some groups and the same number of reservoirs are put parallelly. Each reservoir groups accepts the inputs from the corresponding group and some adjacent grids and predict the dynamics of the corresponding group.

When using the parallelized reservoir computing, the prediction accuracy did not become worse catastrophically even if we reduce half of the observations. This point was slightly different from the results of the original paper. The sensitivity of RC-Obs to the reduction of the observation points is still much greater than that of LETKF-Ext. We have modified some expressions to present the result more precisely.

> Lines 400-406: However, RC-Obs showed a greater sensitivity to the density of observation points than LETKF-Ext. Figures 6a and 6b show the sensitivity of the prediction accuracy of LETKF-Ext and RC-Obs, respectively, to the number of observed grid points. Observation is reduced as uniformly as possible. The choices which grid point to observe is shown in Table 2. Even though we can observe a small part of the system, the accuracy of LETKF-Ext changed only slightly. On the other hand, the accuracy of RC-Obs gets worse when we remove a few observations. As assumed in the section 2.4, we verified that RC-Obs is more sensitive to the observation sparsity than LETKF-Ext.

The other results did not significantly change even though we modified the experiment design using the 40-dimesional L96 model and the parallelized reservoir computing.

*(3.7) I believe that the English of the manuscript could be significantly improved.*
→ We have corrected the points raised by the reviewer.

*As a consequence, I believe the manuscript requires major revisions before being acceptable for publication.*

*2. Typos, remarks and suggestions, some related to the main issues:*

*(3.8) l.46: "have been receiving"→"have received"*

→ We have modified it in the revised version of the paper following the reviewer's instruction.

*(3.9) l.69: Brajard et al. 2020a and Brajard et al. 2020b correspond to the same paper. You can safely remove Brajard et al. 2020a (which is the arXiv preprint of Brajard et al. 2020b). However, there is another 2020 paper from these authors which is directly relevant to your manuscript, see below in the references (Brajard et al.,2020b).*

→ We have deleted Brajard et al., 2020a from the References. Brajard et al.,2020b is taking a similar strategy to ours, in which the machine learning model is trained with the analysis variables of data assimilation methods to interpolate the observation. Considering the studies like Brajard et al., 2020a, Brajard et al., 2020b, Bocquet et al., 2020, Dueben and Bauer, 2018, we believe that the study on combining data assimilation and machine learning is getting popular in NWP and related fields. Brajard et al. 2020b has been cited following the reviewer's instruction in the discussion section, along with the discussion above.

> Lines 500-503: Recently, some studies proposed methods to combine data assimilation and machine learning, to emulate the system dynamics from imperfect model and observations (e.g. Bocquet et al., 2019; Brajard et al., 2020; Bocquet et al., 2020), and these approaches are getting popular. Our study significantly contributes to this emerging research field.

*(3.10) l.71-72: "However, their method needs to iterate the data assimilation and training, which is computationally expensive and infeasible toward the real-world problem.": Not really. Actually it depends on the number of iterations. One can use only one iteration for instance – this is actually what you do. Please mitigate your statement.*

→ Thank you for pointing that out. This statement was written since the surrogate model's error converges after about 20 iterations and the first few iterates did not seem to give accurate prediction compared to the ones after convergence. However, as the reviewer pointed out, one can stop the training with a few iterates, and this also provides the reasonably accurate surrogate model. The statement in the original paper was certainly too strong and we have mitigated it in the revised version of the paper.

> Lines 73-78: However, their method needs to iterate the data assimilation and training until the prediction accuracy of the trained model converges. Although one can stop the iteration

in a few times, it can be longer and the training can be computationally expensive if one should wait until the convergence.

We believe that the first iterate of the method in Brajard et al., 2020 is different from what we did in current study. They assume that one has no information on the state dynamics and the machine learning based surrogate model is used in the data assimilation step instead of the numerical model. However, we have assumed that we have a biased process-based model, and data assimilation is conducted using this model. Since the description of our method may not be sufficient and misleading, we have modified the description of our newly proposed method. See also the response to (3.4).

*(3.11). l.67-75: Bocquet et al. (2020) have recently proposed to use a local EnKF (LETKF for instance) coupled with machine learning. This must be cited since this is very relevant to your paper.*
→ We strongly agree that this paper is quite relevant to our study. They proposed a method to combine EnKF and machine learning methods to obtain both the state estimation and the surrogate model of the system at the same time online. We have cited this study in the introduction section.

Lines 76-78: Bocquet et al. (2020) proposed a method to combine EnKF and machine learning methods to obtain both the state estimation and the surrogate model online. They showed successful results without using the process-based model at all.

Bocquet et al. (2020) is also cited in the discussion section as relevant works on combining data assimilation and machine learning to accurately predict chaotic systems. See also the response to (3.9).

*(3.12) l.94-97: Your definition of the periodic nature of the L96 model is not precise enough, since x0 or xm+1 are not defined for instance. Please improve the wording of your definition.*
→ The periodic nature of L96 is represented in the original manuscript by the word "ring structured" and "$x_m$ is adjacent to $x_1$", but this was not enough to define the all possible variables emerging in equation (1). To make the definition complete, we have modified the wordings in the revised manuscript.

Line 104: where $F$ stands for the force parameter, and we define $x_{-1} = x_{m-1}$, $x_0 = x_m$, and $x_{m+1} = x_1$.

*(3.13) l.100: The characteristic time of the L96 model has been discussed in the original paper by Lorenz and Emanuel (1998). If don't believe that you should cite Miyoshi et al. 2005 here in the text.*
→ As the reviewer pointed out, this point was discussed in detail in the original paper by Lorenz & Emanuel (1998). We have modified the expressions and the citation here in the revised version of paper. Revisions are shown in the response to (1.4) pasted below.
* * *
*(1.4) The strangest thing is the choice of $m = 8$ for the experimental design. The standard L96 with $m = 40$ variables is known to be very chaotic and have 13 positive Lyapunov exponents. In that regime, a lot is known about the system. However as far as the reviewer can tell, the case of $m = 8$ as to be weakly chaotic by necessity. It would be of benefit to the casual reader (and the reviewer) to state some facts about this regime of the system.*

→ As the reviewer pointed out, $m = 8$ is not commonly used in the previous studies. We conducted new experiments with the 40-dimensional L96 system with parallelized reservoir computing which is more suitable for the prediction of higher dimensional systems. We have added the description of the 40-dimensional L96 system introduced by Lorenz & Emanuel (1998).

> Lines 105-110: It is known that the model with $m = 40$ and $F = 8$ shows chaotic dynamics with 13 positive Lyapunov exponents (Lorenz & Emanuel, 1998), and this setting is commonly used in the previous studies (e.g. Kotsuki et al., 2017; Miyoshi, 2005; Penny, 2014; Raboudi et al., 2018). The time width $\Delta t = 0.2$ corresponds to one day in real atmospheric motion from the view of dissipative decay time (Lorenz & Emanuel, 1998).

In the revised version of the paper, all the sentences that refer to the dimensions of state space have been modified, and all results have been updated to the 40-dimension L96 system. Figure 4, 5, 7-10 (Figures 2-7 in the original paper) have been replaced in the revised version of the paper. It should be noted that the primary findings of the original paper do not change when we changed the dimension of the L96 system.

[Figure]

**Figure 4.** The Hovmöller diagram of (a) Nature Run, (b) A prediction of RC-Obs, (c) difference of (a) and (b). Horizontal axis shows the timesteps and vertical axis shows the nodal number. Value at each timestep and node is represented by the color.

[Figure]

**Figure 5**. The *mRMSE* time series of the predictions of RC-Obs with perfect observation. Horizontal axis shows the timestep and vertical shows the value of *mRMSE*.

[Figure]

**Figure 7**. The *mRMSE* time series of the predictions of (a)LETKF-Ext and (b)RC-Obs with spatially sparse observation. Each color corresponds to the number of the observation points.

[Figure]

**Figure 8**. The same as figure4, for the RC-Anl prediction.

[Figure]

**Figure 9**. The *mRMSE* time series of the predictions of (a)LETKF-Ext and (b)RC-Anl with biased model. Each color corresponds to each value of *F* term.

[Figure]

**Figure 10**. The *mRMSE(80)* of the predictions of LETKF-Ext(red) and RC-Anl(blue) for each model bias. Horizontal axis shows the value of the force parameter of equation (1) (8 is the true value) and vertical axis shows the value of *mRMSE*.
* * *
*(3.14) l.108: "and identically distributed on Gaussian distribution"→"and identically distributed from a Gaussian distribution"*

→ We have modified it in the revised version of the paper following the reviewer's instruction.

*(3.15) l.119: "in some countries": this is too vague a statement; which ones for instance?*

→ Scraff et al. (2016) shows the LETKF application to the operational COSMO model, which is used in Germany. We have clarified this point in the revised version of the paper.

> Lines 129-130: LETKF is also used for the operational NWP in some countries (e.g. Germany; Schraff et al., 2016)

*(3.16) l.126: "and the time width of k corresponds to the assimilation window": This statement is not clear to me.*

→ Subscript k here stands for the time step at which estimations are generated, and number of the timesteps between time $k$ and $k+1$ is equal to the assimilation window setting of LETKF. This point was clarified in the revised version of the manuscript.

> Lines 137-138: and the time width between $k$ and $k+1$ corresponds to the assimilation window

In addition, the description of the term "assimilation window" was not precise and sufficient. We have modified the statement in the revised version of the manuscript.

> Lines 132-135: The analysis step makes the state estimation based on the forecast variables and observations. The forecast step makes the prediction from the current analysis variables to the time for the next analysis using the model. The interval for each analysis is called "assimilation window".

*(3.17) l.129: I suggest the change "and N(0,Q) means the Gaussian distribution"→"and N(0,Q) means the multivariate Gaussian distribution" in order to contrast this definition with your previous definition for the univariate N(0,).*

→ We have revised the manuscript according to your identification.

*(3.18) l.134: "extracted from"→"sampled from"*

→ We have modified it in the revised version of the paper following the reviewer's instruction.

*(3.19) l.136-141: It is not clear at this point why you would assume the model to be linear.*

→ We meant that the Gaussian nature of the state variables collapses in a strict sense when the nonlinear transformation is applied (in application, one can assume that the state variables keep being Gaussian random variables after a time evolution of short assimilation window even if the model is nonlinear). The original paper might be confusing since the description of Kalman Filter begins right after this sentence without detail explanations on this point.

The linear assumption here is removed and the discussion written above is added to the revised version of the manuscript.

Lines 142-147 : ==Since the error in the model is assumed to follow the Gaussian distribution, forecasted state $x^f$ can also be considered as a random variable from the Gaussian distribution. When the assimilation window is short, the Gaussian nature of the forecast variables is preserved even if the model dynamics is nonlinear. In this situation, the probability distribution of $x^f$ (and also $x^a$) can be parametrized by mean $\overline{x^f}$ ($\overline{x_k^a}$,) and covariance matrix $P^f$ ($P_k^a$).==

*(3.20) l.149-150: "If either the model operator M or observation operator H is nonlinear, we cannot directly use this method.": that is a bit of an exaggeration. If the model is mildly nonlinear, the extended Kalman filter can be used. Please reformulate.*

→ This statement was indeed too strong considering the applicability of the extended Kalman Filter to weak nonlinear models. However, we have deleted the introduction of Kalman Filter and this description no longer exists in the revised version of the manuscript. See also the response to (3.1).

*(3.21) l.153: "is EnKF. EnKF uses"−→"is the EnKF. The EnKF uses"*

→ This statement was deleted in the revised version of the paper since we removed the description of Kalman Filter following the reviewer's instruction (3.1).

*(3.22) l.161: Your statement is misleading; the covariances matrices in state space are actually never explicitly computed in the (L)ETKF! Please correct this.*

→ We fully agree with this reviewer's comment. In the revised version of the paper, we have modified our description in the following:

Lines 171-186: ==From the forecast ensemble, the mean and the covariance of the analysis ensemble can be calculated in the ensemble subspace as follows:==

$$\overline{w_k^a} = \widetilde{P}_k^a \left( HX_k^f \right)^T R^{-1} \left( y^o - H\overline{x_k^f} \right)$$
$$\widetilde{P}_f^a = \left[ (k-1)I + \left( HX_k^f \right)^T R^{-1} HX_k^f \right]^{-1} \tag{7}$$

where $w_k^a, \widetilde{P}_f^a$ stands for the mean and covariance of the analysis ensemble calculated in the ensemble subspace. ==They can be transformed into model space as follows:==

$$\overline{x_k^a} = \overline{x_k^f} + X_k^f \overline{w_k^a}$$
$$==P_k^a = X_k^f \widetilde{P}_k^a \left( X_k^f \right)^T== \tag{8}$$

==On the other hand, a==s equation (6), we can consider the analysis covariance as the product of the analysis ensemble matrix:

$$P_k^a = \frac{1}{N_e - 1} X_k^a (X_k^a)^T \tag{9}$$

where $X_k^a$ is the matrix whose $i$th column is the variation of the $i$th ensemble member from the mean for the analysis ensemble. Therefore, decomposing $\widetilde{P}_k^a$ of equation (7) into square root, we can get each analysis ensemble member at time $k$ without explicitly computing the covariance matrix in the state space:

$$W_k^a(W_k^a)^T = \widetilde{P}_k^a, \qquad x_k^a = \overline{x_k^f} + \sqrt{N_e - 1}\, X_k^f w_k^a \tag{10}$$

where $w_k^a$ is the $i$th column of $W_k^a$ in the first equation.

*(3.23) l.171, Eq.(10): The second equation is incomplete ;xfk is missing.*
→ We have modified it in the revised version of the paper following the reviewer's instruction.

*(3.24) page 10: How come you don't ever mention localization for the LETKF?*
→ The localization was explained in the section 3 of the original paper. Since the page 10 is the description of "L"ETKF, we should have also added the explanation of localization in this section. We have added the explanation of localization in the section 2 in the revised version of manuscript. See the response to (3.3)

*(3.25) l.180: Why use the dedicated name "LETKF-Ext"? Forecasting from the analysis is just standard proceeding. I may have missed something here.*
→ We agree that forecasting from the analysis is just a standard procedure. However, in the following sections, LETKF is used in a different purpose; to make the analysis time series as the training data for RC-Anl. The name LETKF-Ext was given for the purpose of clarifying this difference. This point has been clarified in the revised version of the manuscript.

> Lines 196-198: This way of making prediction is called "Extended Forecast", and we call this prediction "LETKF-Ext" in this study, to distinguish it from the forecast-analysis iteration of LETKF.

*(3.26). l.184: "shows the architecture." → "shows its architecture."*
→ We have modified it in the revised version of the paper following the reviewer's instruction.

*(3.27). l.185: "in the Section 1, the previous works" →"in Section 1, previous works"*
→ We have modified it in the revised version of the paper following the reviewer's instruction.

*(3.28) l.189: "uk∈Rm": is it m or M as displayed in l.191?*
→ "m" and "M" are both used as the same meaning, the dimension of the state space (this time, 40). We have unified them into m (lowercase) in the revised version of manuscript.

*(3.29) l.194: "is extracted from uniform distribution"→"is sampled from the uniform distribution"*

→ We have modified it in the revised version of the paper following the reviewer's instruction.

*(3.30) l.197: I guess v should be bold, as a vector.*

→ Yes, this is a mistake. We have modified as the reviewer pointed.

*(3.31) l.200: "is the operator for nonlinear transformation."→"is an operator of nonlinear transformations."*

→ We have modified it in the revised version of the paper following the reviewer's instruction.

*(3.32) l.202: "Therefore, the computational cost required to train RC is small": You do not explain why! In particular you do not mention that the problem to optimise is linear (the loss function is quadratic).*

→ The fact that the optimization problem is linear could be found in the equation (15). However, this point was not connected to the computational cost in the original version of the paper, as the reviewer pointed out. We have stressed that the small computational cost is due to this linearity in both introduction and method sections in the revised version of manuscript.

Lines 63-66: RC can learn the dynamics only by training a single matrix ==as a linear minimization problem j==ust once, while other neural networks have to train numerous parameters and need many iterations (Lu et al., 2017).

Lines 220-223: It is important that $A$ and $W_{in}$ are fixed and only $W_{out}$ will be trained by ==just solving a linear problem==. Therefore, the computational cost required to train RC is small and it is an outstanding advantage of RC compared to the other neural network frameworks.

In addition, it is stressed that the problem to calculate is just a linear function.

Lines 231-232: ==Although the objective function (14) is quadratic, it is differentiable, and the optimal value can be obtained by just solving a linear equation as follows:==

*(3.33) l.213: "matrix"→"matrices"*

→ We have modified it in the revised version of the paper following the reviewer's instruction.

*(3.34) l.240-241: It seems to me like the first iteration of Brajard et al. 2020' loop.*

→ We believe that our strategy is conceptually different from the first iterate of the method in Brajard et al., 2020 although we can agree that they are similar. See also the response to (3.10).

*(3.35) l.236-242: What you intend to do is not clear enough to me. Please be more specific. For instance, give the corresponding algorithm. What is the model used by the LETKF supposed to be?*

→The description of the method of RC-Anl was quite insufficient and unclear. In the LETKF step for the RC-Anl, the L96 model with a bias in the Force term was used as the model. These statements have been significantly revised in the revised version of the manuscript, and the detail of the model has also been added. See the response to (3.4).

*(3.36) p.15: A state space dimension of m= 8 is small while the ensemble size Ne= 20 is quite large for such state space size. That makes localization totally unnecessary if not useless. This is really problematic as a demonstration of the method.*

→ We agree that the state space dimension was too small to use no less than 20 ensemble data assimilation, and thus the localization is unnecessary in this case. In order to show the effectiveness of localization and the expandability to higher dimensional systems, we have conducted different experiments with the 40 dimensional L96 system along with parallelized reservoir computing. See the response to (3.13)

*(3.37) l.253: "F term in"→"The F term in"*

→ We have modified it in the revised version of the paper following the reviewer's instruction.

*(3.38) l.253-254: "F term in the model was changed to represent the model bias.": This is only one degree of freedom. Any good data assimilation method can handle this without resorting to machine learning, see for instance Bocquet and Sakov (2013). This undermines your demonstration to some extent.*

→ We fully agree that some of the data assimilation methods can handle the model bias induced by uncertainty in model's parameters. Even if the number of parameters is larger, one can correct them by optimization methods if one identifies the parameters which are the source of model bias. However, when the sources of biases are unknown, the optimization approach is no longer effective, and it is difficult to raise the accuracy of the model by putting the parameters into the state vector of EnKF or EnKS. The motivation of our study is to emulate the system dynamics from the observation and biased model especially when the source of model's bias is unknown. In this problem setting, the number of unknown parameters is not primarily important. This point was indeed unclear in the original version of the paper and has been clarified in the revised version of the manuscript, along with the citation of Bocquet and Sakov (2013).

> Lines 320-327: Here, we assume that the source of the model bias is unknown. When the source of bias is only the uncertainty in model parameters, and uncertain parameters which

significantly induce the model bias is completely identified, optimization methods can estimate the value of the uncertain parameters to minimize the gaps between simulation and observation. This problem can also be solved by data assimilation methods (e.g. Bocquet and Sakov, 2013). However, it is difficult to calibrate the model when the source of uncertainty is unknown. Our proposed method does not need to identify the source of model bias so that it may be useful especially when the source of model bias is unknown. It is often the case in the large and complex model such as NWP systems.

*(3.39) l.262, Eq.(15): the work/concept of "localisation" has not been mentioned once. Has it?*
→ No. The concept of localization was not explained through the section 2, and the description ended up an explanation of EnKF. We have added the explanation of localization in the section 2 and modified the description of this part in the revised version of the manuscript. We have already addressed this point in our responses to the comment (3.2) and (3.22).

*(3.40) l.267: "The configuration of RC used in this study was similar"→"The configuration of RC used in this study is similar"*
→ We have modified it in the revised version of the paper following the reviewer's instruction.

*(3.41) l.267, Eq.(16): The words "odd" and 'even" need not be italicised.*
→ We have modified it in the revised version of the paper following the reviewer's instruction.

*(3.42) From line 271 till the end of the numerical section, please use the present tense, not the past tense.*
→ We have modified it in the revised version of the paper following the reviewer's instruction.

*(3.43) page 16: You could also use an (L)ETKF that accounts for model error, for instance that of Raanes et al. (2015). All NWP data assimilation methods have model error correction steps.*
→ We fully agree that all NWP (ensemble) data assimilation methods have model error correction steps because the background covariance is underestimated without considering model noise. We used the covariance inflation method to maintain the sufficiently large background covariance. Raanes et al. (2015) had the similar motivation to the covariance inflation and compared their proposed method with the covariance inflation. Although we mentioned the covariance inflation in the original version of the paper, it was indeed unclear that we used it to empirically account for model noise. We have clarified this point in the revised version of the paper:

Lines 186-188: A covariance inflation parameter is multiplied to take measures for the tendency of data assimilation to underestimate the uncertainty of state estimate ==by empirically accounting for model noise (see equation (3)).==

Lines 335-338: In equation (10), a "covariance inflation factor", which was set to 1.05 in our study, was multiplied to $\widetilde{P}_k^a$ in each iteration ==to maintain the sufficiently large background error covariance by empirically accounting for model noise (see equation (3)).==

It should be noted that the model noise which we and Raanes et al. (2015) address should be modeled as Gaussian noise with zero mean (see equation (3)) so that the model error correction discussed here cannot directly address the general systematic bias of models.

*(3.44) l.295: What you refer to is the "forecast lead time".*

→ Yes, this is so called forecast lead time. We have added the notes.

Lines 375-377: Using this metric, we can see how the prediction accuracy is degraded as time elapses from initial time ==(so called "forecast lead time").==

*(3.45) l.312: It seems to me that you cannot refer to figures of the supplement material. Your paper has to be self-contained. Please incorporate the figures or remove the paragraph. You may want to discuss this issue with the Editor.*

→ Thank you for pointing that out. We have removed the supplementary materials and corresponding paragraph. We have incorporated Figure S1 in the supplement material as Figure 6 in the revised version of the paper, and we have added detailed description of the figure that is referred to as the supplement material in the original version of the paper.

Lines 394-398: ==Figure 6a and 6b show the effect of the observation error on the prediction skill. The value of observation error== $e$ ==is changed from== 0.1 ==to== 1.5 ==and the== *mRMSE* ==time series is drawn. We can see that LETKF-Ext is more sensitive to the increase of observation error than RC-Obs, although the LETKF-Ext is superior in accuracy to RC-Obs within this range of observation error.==

[Figure]

**Figure 6**. The *mRMSE* time series of the predictions of (a)LETKF-Ext and (b)RC-Obs with noisy observation. Each color corresponds to the observation error indicated by the legend.

*(3.46) l.324: "under imperfect models": do you mean "perfect model"? This is very confusing because you just discussed perfect model experiments and will do so again in the next sentence of the same paragraph (figure 5). It is only in the next paragraph that you report experiments with imperfect model. Please improve the text.*

→ Yes, this paragraph and Figure 5 shows the results with perfect model, and the description here was a mistake. We have modified it in the revised version of the manuscript.

> Lines 408-409: We tested the prediction skill of our newly proposed method, RC-Anl, under perfect models and sparse observations.

*(3.47) l.334-335: To which values do you change F?*

→ F term was moved from 5.0 to 11.0, which can be confirmed in the Figure 6 and 7 in the original manuscript. Since these values cannot be found in the text, we have explicitly mentioned this range in the revised version of the paper.

> Lines 418-420: The $F$ term in equation (1) was changed from the true value 8 (the $F$ value of the model for Nature Run) to values in $[5.0, 11.0]$ as the model bias, and the accuracy of LETKF-Ext and RC-Anl is plotted.

*(3.48) Figure 2: Where are the tags (a), (b), ..., (e) in the panels?*

→ Tags are missing in the original version of paper. We have added them in the revised version.

*(3.49) Figure 4: You could use a log scale for the y-axis (RMSE) to see what happens for the shorter forecast lead times.*

→ We have replaced these figures with the ones showing the 40-dimensional results. We believe that the corresponding figures are now clear to see through the whole forecast lead time without adopting the log scale axis. We have decided not to change this aspect of the paper.

*(3.50) l.336-338 and Figure 6: The comparison is visually difficult. I suggest to merge Figure 6a and 6b which should facilitate the comparison.*
→ We admit that the Figure 6a and 6b (9a and 9b in the revised version of the manuscript) is difficult to compare. However, merging these two figures will yield more complicated figure since the lines are winding here and there. The Figure 7 (10 in the revised version), which is a truncated visualization of the same result, is shown to make the visual comparison easy. We believe that this figure enables us the sufficient comparison and decided not to merge the Figures 6a and 6b.

*(3.51) l. 351: You cannot refer to figures of the supplement material. Your paper has to be self-contained. Please incorporate the figures or remove the paragraph.*
→ We have incorporated the figure as the Figure 11 in the revised version of the manuscript.

[Figure]

**Figure 11**. The *mRMSE* time series of the predictions of RC-Anl with various length of training data, with perfect observation and perfect model. Each color corresponds to the value of the size of training data.

*(3.52) l.388-390: No, in their method one can limit the number of iterations, to 1 for instance, just as you do. These authors also have a sequel to this paper, where they use a physical imperfect model as a first iteration, just as you do and this is potentially applicable to high-dimensions (Brajard et al., 2020b). By the way, you are using m=8 while they are using m=40, so they are a little closer to high-dimension than you are.*

→ In the original version of the paper, our explanation of the applicability of Brajard et al., 2020 was misleading. We have already addressed this issue in the introduction section as our response to the comment (3.10). We have also modified the description here in the revised version of the manuscript.

> Lines 468-472: Brajard et al. (2020) ==iterated== the learning and data assimilation until they converge, because it replaced the model used in data assimilation with CNN. Although their model-free method has an advantage that it was not affected by the process-based model's reproducibility of the phenomena, it ==can be== computationally expensive ==since the number of iterates can be relatively large==.

However, we believe that our method is different from the first iterate of Brajard et al. (2020). See our response to (3.10).

The reviewer mentioned that Brajard et al. (2020) set m = 40 so that they are better toward the application of high-dimension systems. We have also set m = 40 in the revised version of the paper and obtained the robust results. Therefore, we believe that this point has been addressed in the revised version of the paper. See our response to the comment (3.13).

*(3.53) l.390: "contrary"→"By contrast"*
→ We have modified this in the revised version of the manuscript.

*(3.54) l.406: "The advantage of our proposed method is that we allow both models and observation networks to be imperfect.": So do a lot of other methods that use other ML technique than RC (Bocquet et al., 2019; Brajard et al., 2020a; Bocquet et al., 2020) for instance.*
→ We intended to show the superiority of our method over the series of RC studies (not works raised by reviewers). We agree that there are some other studies that accounts for the uncertainty in both models and observations. We realized that this sentence was misleading and we have clarified that our proposed method addressed the current issue of the series of RC studies in the revised version of the paper.

> Lines 498-500: The advantage of our proposed method ==compared to these RC studies== is that we allow both models and observation networks to be imperfect.

Moreover, other studies that trained ML with imperfect models and observations are cited right after this sentence to show that our study is not the only. See the response to (3.9)

*(3.55) l.413-416: When reading the abstract of your paper, I honestly thought that you would use the local RC, since you were claiming to use the LETKF. Doing so would have make your paper much stronger and more consistent (using a local RC with a local ETKF). Why did you not try it?*

→ We have conducted the experiments with parallelized RC and obtained similar results. See also the response to (3.6).

*(3.56) l.421-422: "These results imply that our proposed method can be applicable to various realistic problems.": Testing the method on a 8-dimensional problem with a global RC does not make it applicable to various realistic problems! Please mitigate this too bold statement.*

→ We have conducted additional experiments with the 40-dimensional L96 system, along with localization which is a strong tool for the application to higher dimensional problems. The new results seem to make this statement more convincing. However, we agree that it is still too strong. We have mitigated it in the revised version of the manuscript. The comment (1.8) is similar to this comment so that we have attached the response to the comment (1.8) below. Please find how we addressed the reviewer's concern.
* * *
*(1.8) In section 5 "Discussion" the authors mention the generalization of their methodology to other dynamical systems. The authors focus on the generalization to "higher dimensional systems" as their main worry. The reviewer believes that another more substantial worry is that Lorenz '96 is known to be ergodic, while most systems used for NWP are possibly not, (the authors mention the KS system, but that is also possibly ergodic for well-posed initial conditions). It would be of benefit to mention other possible shortcoming on testing only on the Lorenz '96 system.*

→ We agree that the main concern of testing only on the L96 is that this model is ergodic. However, it is known that the 40-dimensional L96 shows strong chaos and large Lyapunov exponents, and this model has been used as a good initial testbed toward the application to large and non-ergodic NWP problems. In the revised version of the manuscript, the scalability in terms of dimensionality and ergodicity are discussed separately, and main shortcomings on testing only on the L96 system are explicitly mentioned.

Lines 505-508: Although we tested our method only on 40-dimensional Lorenz 96 system, Pathak, Hunt et al. (2018) indicated that parallelized RC can be extended to predict the dynamics of substantially high dimensional chaos such as 200-dimensional Kuramoto-Sivashinski equation with small computational costs. It implies that the findings of this study can also be applied to higher dimensional systems.

Lines 515-518: However, since Lorenz 96 model (and other conceptual models such as Kuramoto-Sivashinski equation) are ergodic, it is unclear that our method can be applied to real NWP problems directly, which are possibly non-ergodic.

==Although our proposed method has a potential to extend to larger and more complex problems,== ==further studies are needed==.
* * *
*(3.57) l.431-433: "Our new method is robust to the imperfectness of both models and observations so that it is feasible to apply it to the real NWP problem.": I really don't believe you can make such statement, from an 8-dimensional L96 model. Please remove that statement, which could be shocking to many colleagues working in numerical weather prediction and data assimilation.*

→

We agree that the statement in the original paper was too strong. We have removed the statements on the real NWP problems, and modified following the revision in the response to (3.56).

Lines 527-529: Our new method is robust to the imperfectness of both models and observations ==and we might obtain similar results in higher dimensional and more complexed systems.==

---

## Author Response (AR2)

Response letter of gmd-2020-211

Dear Editor,

Please find the revised version of our manuscript "Combining Ensemble Kalman Filter and Reservoir Computing to predict spatio-temporal chaotic systems from imperfect observations and models", which we would like to resubmit for publication in *Geoscientific Model Development.*

Comments made by the reviewers were highly insightful. They allowed us to greatly improve the quality of the manuscript. We described the responses to the comments.

Each comment made by the reviewers is written in *italic* font. We numbered each comment as (n.m) in which n is the reviewer number and m is the comment number. In the revised manuscript, changes are highlighted in yellow in the single-column and double-spaced paper.

We trust that the revisions and responses are sufficient for this manuscript to be published in *Geoscientific Model Development*.

Sincerely,
Futo Tomizawa, Yohei Sawada

**Responses to the comments of Topical Editor**

*(0.1) There has been a lot of recent activity in using ML for data assimilation; a more thorough review would help the authors better position this work in the current context. A few possible suggestions:*

*10.1007/s12551-020-00776-4*

*10.1137/20M1349965*

*10.1038/s42254-021-00314-5*

→ Recent activity in using ML for NWP tasks was introduced in Section 1, but it was not discussed in the context of the general dynamical system theory. We cited more papers including ones proposed by the editor to reinforce our discussion.

> Lines 47-50: On the other hand, model-free prediction methods based on machine learning have received much attention recently. ==In the context of dynamical system theory, previous works have developed the methods to reproduce the dynamics by inferring it purely from observation data (e.g., Rajendra & Brahmajirao, 2020), or by combining a data-driven approach and physical knowledge on the systems (Karniadakis et al., 2021). In the NWP context,== many previous studies have successfully applied machine learning to predict chaotic dynamics.

> Lines 512-514: ==As in the review by Karniadakis et al. (2021), methodologies to train the dynamics from noisy observational data by integrating data and physical knowledge are attracting attentions.== In the NWP context, some studies proposed methods to combine data assimilation and machine learning to emulate the system dynamics from imperfect model and observations (e.g. Dueben & Bauer, 2018; Bocquet et al., 2019; Brajard et al., 2020; Bocquet et al., 2020), and these approaches are getting popular.

However, we have decided not to cite the second paper in the suggestion. It is a proposal of a novel data assimilation method, and it has significantly contributed to the data assimilation research. On the other hand, our proposed method in this work does not depend on specific data assimilation methods and many other methods including 4D-VAR or particle filter can be used instead of LETKF. Thus, we think that the paper is not significantly related to our paper.

**Responses to the comments of Referee #4**

*(4.1) The construction and training of RC (and indeed of any NN) where the number of inputs/outputs equals the number of states of a geophysical model raises the question of dimensionality. The strategy works for toy models like L96, but does it scale to models with millions/billions of variables? The authors should discuss the scalability of the approach.*

→ Previous studies such as Pathak, Hunt et al. (2018) showed that the parallelized reservoir computing can predict the dynamics of Kuramoto-Sivashinski system of up to 200 grid points with tractable computational cost. They also discussed the scalability of the parallelized reservoir computing scheme and applicability to the realistic NWP problems in the sequel study (Wikner et al., 2020). Although we discussed this point in the previous version of the manuscript, we have strengthened the discussion citing the latter paper.

> Lines 522-523: Although we tested our method only on 40-dimensional Lorenz 96 system, Pathak, Hunt et al. (2018) indicated that parallelized RC can be extended to predict the dynamics of substantially high dimensional chaos such as 200-dimensional Kuramoto-Sivashinski equation with small computational costs. Moreover, the applicability to the realistic NWP problems has also been discussed by their sequel study (Wikner et al., 2020). These studies imply that the findings of this study can also be applied to higher dimensional systems.

*(4.2) RC-Obs, equation (17), needs to be explained in more detail. The training equation (14) seems to imply that the inputs and the outputs of the RC live in the same space. This is not the case with the first equation in (17) predicting X^f_{K+1}.*

→ As the reviewer pointed out, the inputs and outputs of the RC must be in the same space, otherwise the prediction accuracy is degraded. Thus, in the equation (17), $x^f_{k+1}$ has the same dimensionality as $y^0_k$, meaning that we can only predict the future state of the observable grid points. Since this point was not clarified in the previous version of the manuscript, we have added some descriptions explaining this point.

> Lines 257-259: As in equation (14), input and output of RC must be in the same space. Therefore, in this case, prediction variables $\boldsymbol{x}^f_k$ has the same dimensionality as $\boldsymbol{y}^0_k$, and the non-observable grid points are not predicted by this prediction scheme.

*(4.3) Depending on how state variables are split, the parallelization of RC can be viewed as a form of localization, where the dynamics of groups of variables in the proximity of each other is trained separately from other groups. The authors need to explain this connection better, and eventually comparer against the predictions of a non-parallel RC.*

→ We agree that the analogy between parallelization of RC and localization is not sufficiently discussed. We added the discussion on that analogy and then highlighted the difference between parallelized RC and ordinary RC.

> Lines 281-286: The strategy of parallelization is similar to the localization of data assimilation. As LETKF ignores correlations between distant grid points, parallelized reservoir computing assumes that the state variable of a grid point at the next time step depends only on the state variables of neighboring points. In contrast, ordinary RC assumes that the time evolution at one grid point is affected by all points in the state space, which may be inefficient in many applications in geoscience such as NWP.

*(4.4) For eqn. (22), should the estimation depend on the initial state as well?*
→ No, the estimation does not necessarily depend on the initial state. Even if we start the data assimilation procedure from a random initial value, we can obtain a good state estimation time series by truncating the head of the analysis time series by the appropriate amount as the "spin-up" duration. We have decided not to change the paper responding to this comment.

*(4.5) Is the nonlinear output function f(r_k) in eqn (25) defined specifically for Lorenz96?*
→ No, similar kinds of nonlinear transformation function is found to be suitable for other spatio-temporal chaotic systems such as Lorenz 63, Kuramoto-Shivasinski, and so on. (e.g., Pathak et al., 2017) On the other hand, other kinds of nonlinear function can also be suitable for the prediction of Lorenz 96 system (e.g., Chattopadhyay et al., 2019). We have clarified this point in the revised version of the manuscript.

> Lines 359-361: Note that the form of the transformation function can be flexible; one can use a different form of the function to predict Lorenz 96 (Chattopadhyay et al., 2019), or the same function can be used to predict other systems (Pathak et al., 2017).

*(4.6) The language can be polished for a better presentation. For example, at line 231: "Although it is quadratic … it is differentiable" does not seem right. Please check throughout*
→ Thank you for pointing that out. We checked the whole manuscript and revised the expressions.

> Line 234: Since the objective function (14) is quadratic, it is differentiable. The optimal value can be obtained by just solving a linear equation as follows: